# Transcriptomic atlas of midbrain dopamine neurons uncovers differential vulnerability in a Parkinsonism lesion model

Behzad Yaghmaeian Salmani[1]*, Laura Lahti[1], Linda Gillberg[1], Jesper Kjaer Jacobsen[1,2], Ioannis Mantas[3], Per Svenningsson[3], Thomas Perlmann[1]*

[1]Department of Cell and Molecular Biology, Karolinska Institutet, Stockholm, Sweden; [2]Department of Neurology, Karolinska University Hospital, Stockholm, Sweden; [3]Department of Clinical Neuroscience, Karolinska Institutet, Stockholm, Sweden

**\*For correspondence:**
behzad.yaghmaeian.salmani@ki.se (BYS);
thomas.perlmann@ki.se (TP)

**Competing interest:** The authors declare that no competing interests exist.

**Abstract** Midbrain dopamine (mDA) neurons comprise diverse cells with unique innervation targets and functions. This is illustrated by the selective sensitivity of mDA neurons of the substantia nigra compacta (SNc) in patients with Parkinson's disease, while those in the ventral tegmental area (VTA) are relatively spared. Here, we used single nuclei RNA sequencing (snRNA-seq) of approximately 70,000 mouse midbrain cells to build a high-resolution atlas of mouse mDA neuron diversity at the molecular level. The results showed that differences between mDA neuron groups could best be understood as a continuum without sharp differences between subtypes. Thus, we assigned mDA neurons to several 'territories' and 'neighborhoods' within a shifting gene expression landscape where boundaries are gradual rather than discrete. Based on the enriched gene expression patterns of these territories and neighborhoods, we were able to localize them in the adult mouse midbrain. Moreover, because the underlying mechanisms for the variable sensitivities of diverse mDA neurons to pathological insults are not well understood, we analyzed surviving neurons after partial 6-hydroxydopamine (6-OHDA) lesions to unravel gene expression patterns that correlate with mDA neuron vulnerability and resilience. Together, this atlas provides a basis for further studies on the neurophysiological role of mDA neurons in health and disease.

## eLife assessment

This **important** study investigated transcriptional profiles of midbrain dopamine neurons using single nucleus RNA (snRNA) sequencing. The authors found more nuanced subgroups of dopamine neurons than previous studies, and identified some genes that are preferentially expressed in subpopulations that are more vulnerable to neurochemical lesions using 6-hydroxydopamine (6OHDA). The results are **convincing** and provide critical information on the heterogeneity and vulnerability of dopamine neurons which will be a foundation for future studies.

## Introduction

Dopamine (DA) neurons localized in the ventral midbrain (mDA neurons) comprise the brain's largest and most extensive source of DA neurotransmission (*Björklund and Dunnett, 2007*). These neurons are known to perform a wide variety of functions, and distinct mDA neuron assemblies are known to play diverse functional roles. For example, mDA neurons of the substantia nigra pars compacta (SNc) are mainly associated with controlling motor functions. In contrast, those in the ventral tegmental

area (VTA) and retrorubral field (RRF) have been associated more strongly with motivation, learning, and cognition (*Björklund and Dunnett, 2007*). Notably, mDA neurons are also characterized by their varying impact on disease. For example, the hallmark motor symptoms of Parkinson's disease (PD) result from the degeneration of SNc neurons. On the other hand, mDA neurons in the VTA have been linked to neural pathways that are affected in addiction, schizophrenia, attention deficit hyperactivity disorder, and other psychiatric conditions (*Björklund and Dunnett, 2007*; *Garritsen et al., 2023*; *Poulin et al., 2020*).

The emergence of single-cell RNA sequencing and other methods that resolve differences in, for example chromatin state between single cells, have increased our understanding of mDA neuron diversity and innervation patterns (*Garritsen et al., 2023*; *Poulin et al., 2020*; *Poulin et al., 2018*). This diversity is of great significance in normal brain functions and disease, for example in forming multiple DA-controlled circuits enabling versatile DA neurotransmission and explaining different vulnerabilities of distinct groups of mDA neurons. However, methods dependent on capturing certain cell types, such as neurons sensitive to cell dissociation, can result in biased datasets where the most sensitive neuron groups are underrepresented. In addition, as the scRNAseq studies of mDA neurons published so far have analyzed a relatively low number of mDA neurons, the resolution of these datasets is still quite limited. Single-nucleus RNA sequencing (snRNA-seq) is a more efficient and reliable sampling alternative for individual neurons (*Krishnaswami et al., 2016*). Indeed, snRNAseq of human mDA neurons, enriched by sorting for NR4A2-labelled nuclei, yielded a 180-fold increase in absolute neuronal numbers compared to similar datasets based on scRNAseq (*Kamath et al., 2022*).

Our present study combined fluorescent sorting and snRNA-seq to explore the diversity of a large number of mouse mDA neurons, resulting in a comprehensive atlas of mDA neuron populations. By additionally analyzing nuclei from a partial toxin-induced mDA neuron degeneration model, we were able to investigate questions of selective and variable vulnerability among these distinct populations. Our findings provide a foundation for future investigations into the projections and functions of molecularly distinct mDA neuron groups.

## Results
### Single nuclei RNA sequencing of the ventral midbrain
Perhaps partly because of the challenges in dissociating and isolating whole mDA neurons, previous studies have only analyzed a relatively small fraction of all mDA neurons from each dissected brain (*Dougalis et al., 2012*; *Hook et al., 2018*; *La Manno et al., 2016*; *Poulin et al., 2014*; *Tiklová et al., 2019*). This raises concerns about the potential for sampling bias in those studies. To overcome this limitation, we used single nuclei RNA sequencing (snRNA-seq) for mice expressing nuclear mCherry under the control of the *Slc6a3* (DAT, dopamine transporter) promoter (*Figure 1A–C*; *Hupe et al., 2014*). Nuclei isolated by fluorescent sorting were sequenced using the 10 X Genomics Chromium platform (10Xv3). Using this strategy, more than 20% of all mDA neurons were consistently collected and sequenced from each brain, an improvement of over 50 X from our previous study (*Tiklová et al., 2019*).

In addition to assessing the diversity among the mDA neuron population, we also aimed to analyze how diverse mDA neuron groups were affected by cellular stress. Accordingly, mice (n=6) were unilaterally injected with 6-hydroxydopamine (6-OHDA, 0.7 µg or 1.5 µg) into the medial forebrain bundle, which leads to partial cell death of mDA neurons on one hemisphere of the brain (*Figure 1A*). To determine differential survival among diverse mDA neurons, nuclei from intact and lesioned hemispheres were collected six weeks after lesioning when the acute phase of cell death had already occurred. To avoid omitting low red fluorescent protein (RFP)-expressing nuclei from analysis, a relatively relaxed gating in fluorescently activated nuclear sorting (FANS) was used, resulting in an approximately equal number of mCherry$^+$ and mCherry$^-$ nuclei from each sample per brain (*Figure 1—figure supplement 1*). After snRNAseq and quality control, 36,051 nuclei from the intact and 32,863 from the lesioned hemisphere were retained for bioinformatic analysis (see Materials and methods, *Figure 1—figure supplement 2*).

Sequencing data were subjected to normalization, variance stabilization, and dimensional reduction and then visualized by uniform manifold approximation (UMAP; *Figure 1D*). Louvain clustering was adopted, dividing all nuclei into 26 clusters (Materials and methods, *Figure 1—figure*

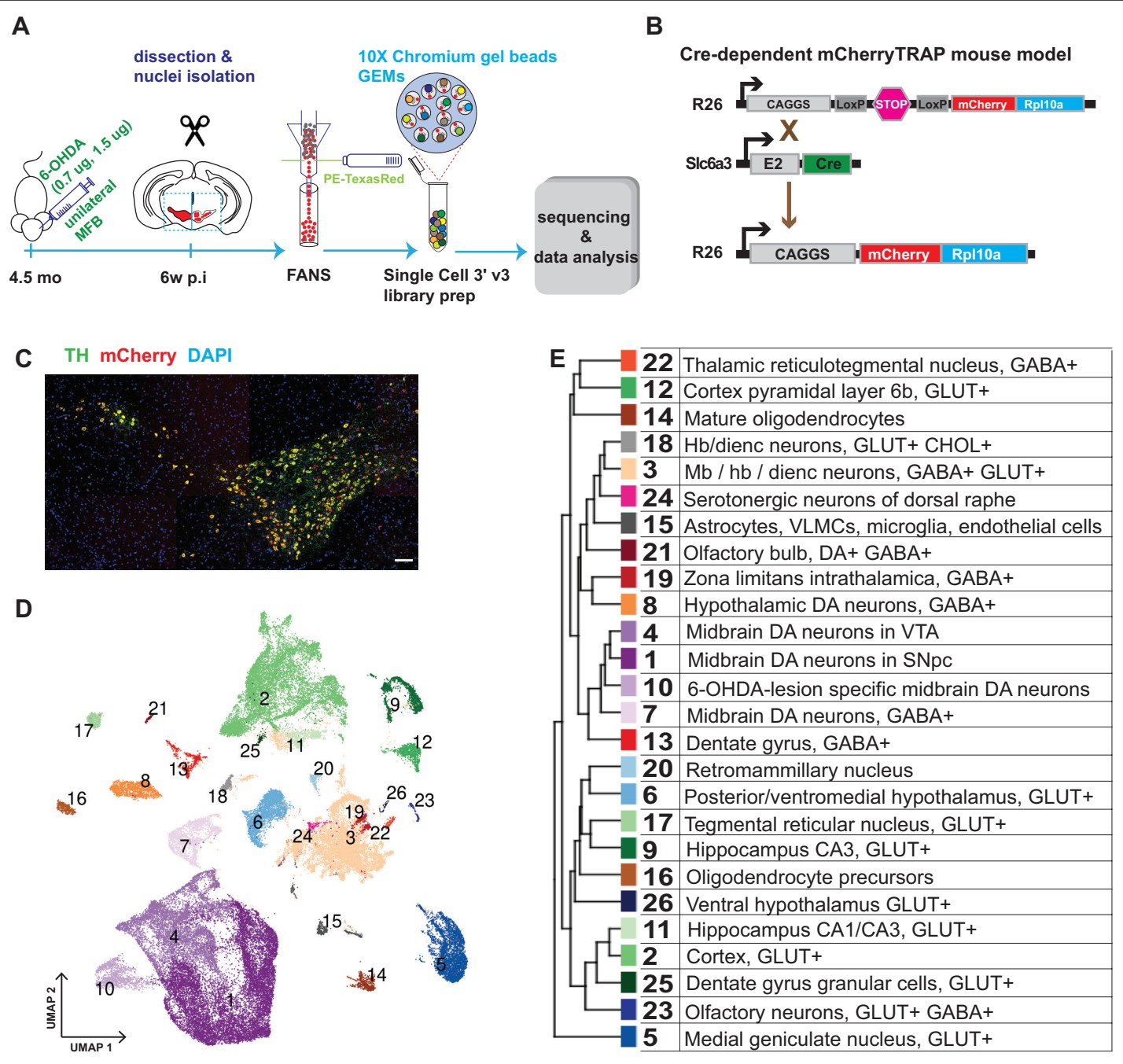

**Figure 1.** Generation of a snRNAseq dataset from the mouse midbrain. (**A**) Schematic design of the study, unilateral injection of low doses of 6-OHDA (0.7, 1.5 µg/µl) in the medial forebrain bundle (MFB), followed by dissection, nuclei isolation and enrichment via FANS, before library preparation, sequencing, and data analysis. (**B**) Transgenic, Cre-dependent mCherryTRAP mouse model. *Slc6a3^{Cre}* and *TrapC^{fl}* lines were intercrossed to generate *Slc6a3^{Cre/+}; TrapC^{fl/fl}* mice. (**C**) Immunohistochemical staining of the mouse ventral midbrain coronal section, showing the overlap of TH and mCherry. Scale bar = 100 µm. (**D**) UMAP projection of the all-nuclei dataset, with cluster color-coding. (**E**) Hierarchical dendrogram of the identified clusters and their descriptions with the same color-coding.

The online version of this article includes the following figure supplement(s) for figure 1:

**Figure supplement 1.** FANS plots for intact, lesioned, and untreated nuclei sorting.

**Figure supplement 2.** Quality control of snRNAseq data.

**Figure supplement 3.** Spatial mapping of main non-mDA-clusters.

*supplement 3*). A hierarchical clustering dendrogram illustrating the relationship between clusters was also constructed, and cluster-specific markers were identified and used to annotate previously known cell types (*Figure 1D and E*; *Figure 1—figure supplement 3*, *Supplementary file 1*). This analysis revealed the presence of different cell types, including oligodendrocytes; oligodendrocyte precursor cells; astrocytes; and various types of neurons known to be localized in the midbrain but also with some inclusion of cell types known to be localized both in the forebrain and hindbrain. Neuronal clusters were mainly comprised of different types of glutamatergic (GLU), gamma-aminobutyric acid (GABA), and dopamine neurons (*Figure 1—figure supplement 3*). This dataset is accessible via our lab homepage: https://perlmannlab.org/resources/.

## Dopamine neuron diversity

Plotting known markers for mDA neurons in the UMAP revealed that a substantial number of all analyzed nuclei, clusters 1, 4, 7, and 10, were derived from mDA neurons (*Figure 1—figure supplement 3*). To focus the analysis on dopamine neurons, 33,052 nuclei expressing above threshold levels of either *Th* or *Slc6a3* ($Th^+$/ $Slc6a3^+$) were selected for further computational analysis. Most of these filtered nuclei, visualized in a de novo UMAP, were positive for several genes known to be expressed in mDA neurons (*Figure 2A and B*). This dataset is accessible via our lab homepage: https://perlmannlab.org/resources/. Included in this analysis are neurons that have been described before expressing *Th* mRNA but no protein (*Morales and Margolis, 2017*). Indeed, our histological analysis showed that although TH protein and mRNA were detected in nearly all mCherry$^+$ cells throughout the midbrain, a few scattered cells were seen that were either mCherry$^+$ but lacked TH protein or, in very rare cases, both mRNA and protein. In addition, we detected some mCherry$^-$ cells that contained only *Th* mRNA (*Figure 2—figure supplement 1*). As our analysis was based on mRNA and not protein expression, these cells were also included in our dopaminergic subset.

K-means clustering, followed by hierarchical clustering, was used to divide $Th^+$/$Slc6a3^+$ nuclei into 71 unique clusters whose relationship was also visualized in a dendrogram (*Figure 2C*). Although the analysis of gene expression in a dot-plot confirmed that most nuclei were dopaminergic, a smaller number of nuclei distributed in clusters expressing low levels of mDA neuron markers showed characteristics of GABAergic, glutamatergic neurons, or maturing myelinating oligodendrocytes (mODC). A few clusters remained unassigned (*Figure 2A and C*, *Supplementary file 2*). Almost all clusters expressing genes required for synthesis and handling DA (dopaminergic) also expressed mDA neuron markers, including *En1*, *Nr4a2*, and *Pitx3*, the latter at a lower abundance, except the ones that expressed hypothalamic dopaminergic markers, including, *Satb2, Prlr,* and *Gad1* (*Figure 2A and C*; referred to as HyDA neurons). The mDA neuron clusters form a major separate branch in the dendrogram, while HyDA neurons constitute a more distant branch (*Figure 2C*). Two closely related clusters (#12 and #26) form a distinctly separate group, predominantly (~95%) composed of nuclei from the lesioned hemisphere. Upon closer inspection, they share the same parent node with mDA clusters marked by high abundance and expression level of *Th*. However, the expression of *Th* and other dopaminergic markers in these two lesion-specific clusters is relatively low (*Figure 2C*). The lesion-specific clusters and the consequences of lesioning will be further described below.

## mDA neuron territories and neighborhoods

Illustration of diversity among different sub-categories of cells, such as mDA neurons, using a dendrogram can give a somewhat misleading impression that individual branching endpoints (referred to as 'leaf nodes') would represent transcriptionally distinct cellular subtypes. However, it may be more accurate to consider differences between closely related mDA neuron groups as a gradually shifting continuum within a gene expression landscape. Thus, we found using a hierarchical definition of the relationship more accurate when using the dendrogram to describe diversity. By this definition, mDA neurons are distinct and constitute a neuron type, clearly different from, for example, HyDA and other types of GABAergic and glutamatergic neurons. Within individual neuron types, such as the closely related mDA neurons, we define the next hierarchical level as 'territories', while 'neighborhoods' is the most highly resolved level of diversity that, based on the bioinformatic analysis, can be defined meaningfully. According to this classification, each one of the territories and neighborhoods should constitute specified adjacent leaf nodes of the cluster tree with a common ancestor node in the hierarchical dendrogram. Notably, leaf nodes of a neighborhood or territory need not be sibling nodes.

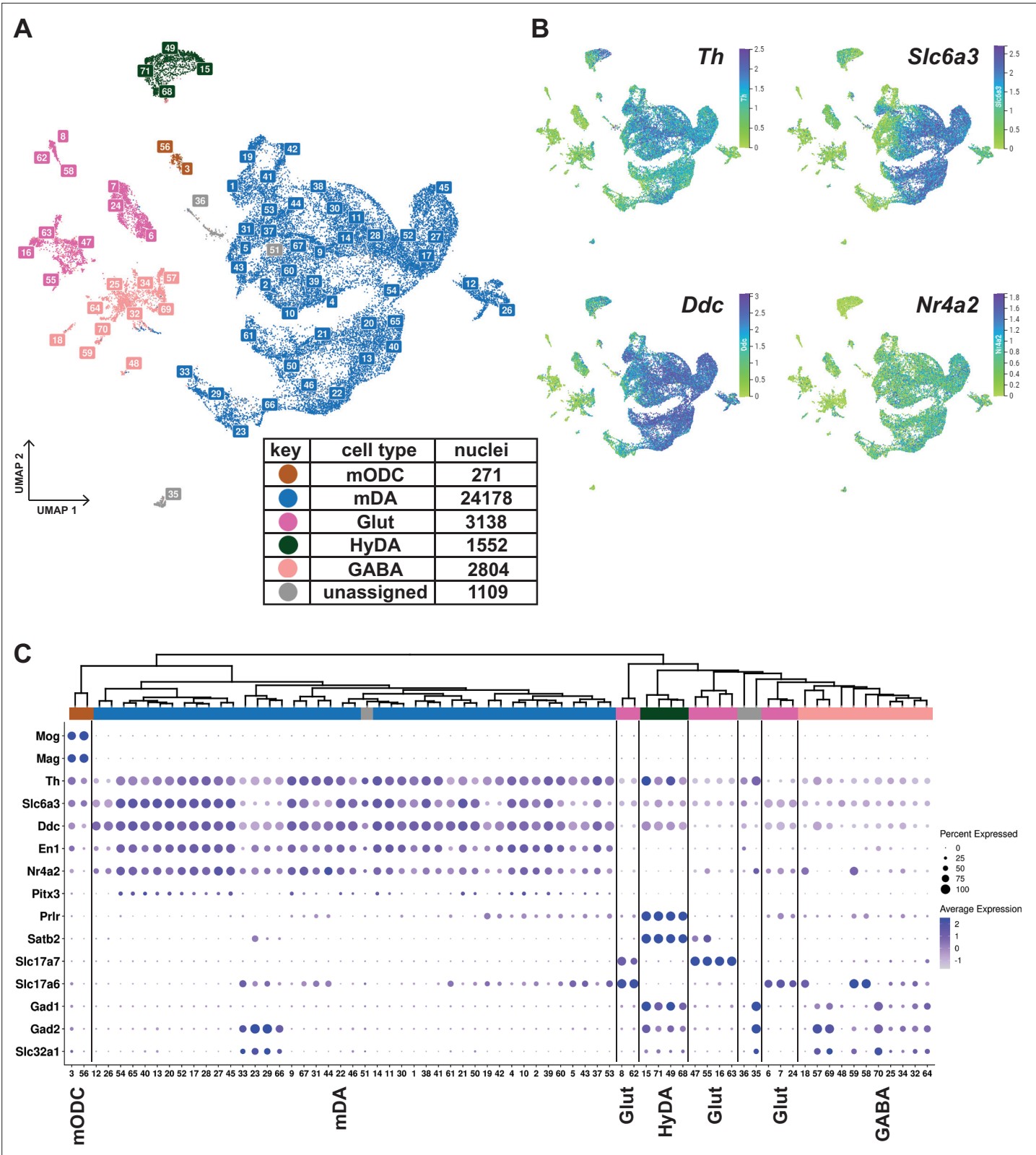

**Figure 2.** Midbrain DA dataset, cell types and composition. (**A**) UMAP projection of the dopaminergic dataset, with numerical labels of the 71 clusters and color-coded for the major cell types. (**B**) Typical dopaminergic markers plotted in UMAP separately, with the top 1 percentile of the normalized gene expression range clipped for better visualization. (**C**) Hierarchical dendrogram of clusters from the dopaminergic nuclei dataset, with the cell-type enriched markers as a dotplot and cell-type color-coding. mODC = mature oligodendrocyte, HyDA = hypothalamic dopaminergic.

*Figure 2 continued on next page*

*Figure 2 continued*

The online version of this article includes the following figure supplement(s) for figure 2:

**Figure supplement 1.** Distribution of *Th* mRNA and protein in different mDA populations.

---

Furthermore, territories and neighborhoods should be distinguishable by clearly defined gene expression signatures and occupy a relatively well-defined space, that is not scattered, in the UMAP. This classification subdivided mDA neurons into 7 territories and 16 neighborhoods (*Figure 3A and C*; *Figure 3—figure supplement 1*). The territories were named according to selected distinguishing genes (*Sox6, Gad2, Fbn2, Pcsk6, Pdia5, Ebf1,* and *Otx2*) and can be identified by additional markers as indicated in *Figure 3* (see also *Figure 3—figure supplement 1*). It should be noted that these markers were typically not unique for a particular territory but, in combination, defined gene signatures that distinguished territories from each other.

The transcription factor Sox6 is known to be expressed in mDA neurons of the SNc and in some neurons the VTA (*Pereira Luppi et al., 2021*; *Panman et al., 2014*; *Poulin et al., 2014*). The *Sox6*-territory included approximately one-third of all sequenced mDA neuron nuclei (*Figure 3A*). This territory expressed other previously defined SNc markers, including *Kcnj6 (Girk2)* (*Figure 3—figure supplement 1*) and additional genes representing a gene signature that included *Col25a1* and *Tigar*, as well as *Nwd2, Pex5l, Cdh11,* and *Serpine2* (*Figure 3C*; *Figure 3—figure supplement 1*). In contrast, expression of *Calb1*, encoding the known VTA marker Calbindin 1, was either absent or was expressed at low levels in *Sox6*-territory nuclei (*Figure 3C*). This almost non-intersecting, mutually exclusive expression of *Sox6* and *Calb1* defined a striking major division among all analyzed mDA neuron nuclei, following the neuroanatomical division of mDA neurons into SNc and VTA (*Figure 3B*).

The remaining six territories consisting of approximately 15,000 mDA nuclei were more diverse and, based on the prominent expression of *Calb1* and absence of high *Sox6* expression, presumably represented mDA neurons localized outside SNc. Indeed, these territories expressed additional VTA markers, including *Otx2* (*Figure 3C*). Expression of mDA neuron markers such as *Th, Ddc,* and *Slc6a3* was particularly weak in the *Gad2*-territory, which displayed typical features of GABAergic neurons, including *Gad2* and *Slc32a1*.

Our next focus was to describe mDA neuron diversity within territories by dividing them into more highly resolved neighborhoods (NHs, *Figure 3C*, *Figure 3—figure supplement 1*). In rare cases, the smallest hierarchical domain for a neighborhood can be one dendrogram leaf node. Still, most consist of at least two leaf nodes. Four neighborhoods were defined within the *Sox6*-territory, two of which expressed high levels of *Aldh1a1*, a previously recognized marker for some SNc neurons (NH1 and NH4). *Sox6*$^+$ + were also distinguished by, *Vcan* and *Col11a* (NH1 and NH2), *Anxa1* and *Fam19a4* (NH1 and NH4), *Grin2c, Ndnf* and *Vill* (NH3 and NH4), and *Aldh1a7* (NH4), among other markers.

The other six territories were each divided into two neighborhoods (*Figure 3C*, *Figure 3—figure supplement 1*). *Gad2*_NH1 expressed *Chrm2* and *Zfp536*, whereas *Gad2*_NH2 expressed *Megf11, Zeb2,* and *Met*. *Fbn2*_NH1 was specified by *Rxfp1* and *Hs3st2* while *Fbn2*_NH2 by *Col23a1, Dsg2* and *Ism1*. *Pcsk6*_NH1 was positive for *Cald1, Pde11a* and *Cpne2*. *Pcsk6*_NH2 contained *Tacr3*$^+$ *Aldh1a1*$^-$ cells, which also expressed *Igf1*. There were also *Tacr3*$^+$ *Aldh1 a1*$^+$ cells, but they belonged to *Otx2*_NH1, which also expressed *Grp, Eya1,* and *Plekhg1*. *Otx2*_NH2, which was low for *Slc6a3*, was identified by *Csf2rb2, Nfib,* and *Nfia*. The *Pdia5*_NH1 expressed *Jph1 Cpne5,* and *Pdia5*_NH2 was positive for *Npy1r, Postn,* and *Wnt7b*. The *Ebf1*_NH1 expressed *Col24a1*, whereas *Ebf1*_NH2 was positive for *Vip*.

## Anatomical distribution of diverse mDA neurons

Next, we investigated the anatomical distribution of the seven mDA neuron territories and their neighborhoods in the adult mouse midbrain. We based our spatial mapping on Allen Mouse Brain Atlas (mouse.brain-map.org) and on our own in situ hybridization and immunohistochemical analyses. In situ hybridization experiments used single probes on individual sections combined with immunohistochemistry. We also used an in situ sequencing strategy that included 49 selected probes, hybridized simultaneously on seven sections across the entire midbrain (*Figure 4B–E*, *Figure 4—figure supplements 1–3*, *Supplementary files 3 and 4*; *Gyllborg et al., 2020*; *Krzywkowski et al., 2017*; *Qian et al., 2020*). Coronal in situ Allen Mouse Brain Atlas data (atlas.brain-map.org) was used as a guide when relevant data were available. For the anatomical localization of territories and neighborhoods,

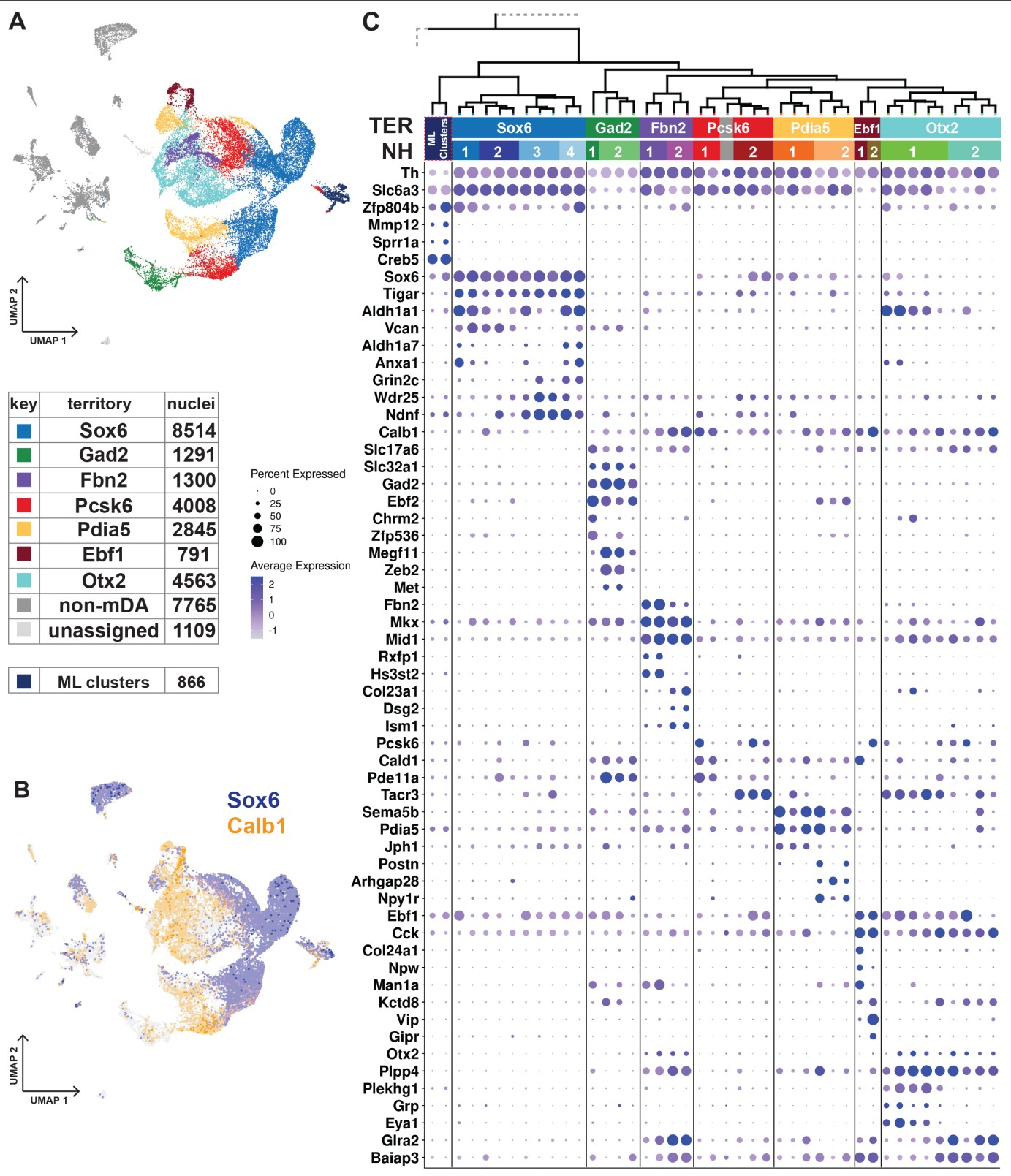

**Figure 3.** Dopaminergic territories and neighborhoods. (**A**) UMAP projection of the dopaminergic dataset with territory color-coding and size. (**B**) Co-expression of *Sox6* and *Calb1* plotted in UMAP, roughly delineating SNc and VTA respectively. (**C**) Hierarchical dendrograms with labels and color-coded territories and their respective neighborhoods. Dotplot shows territory (TER) and neighborhood (NH) specific markers.

*Figure 3 continued on next page*

*Figure 3 continued*

The online version of this article includes the following figure supplement(s) for figure 3:

**Figure supplement 1.** Territory and neighborhood markers of mDA dataset.

we mainly followed Paxinos and colleagues (*Fu et al., 2012*) and the coronal reference atlas from Allen Mouse Brain Atlas. Together these analyses provided the basis of the schematic in *Figure 4F*, showing the distribution of different territories in rostrocaudal levels of the midbrain.

Consistent with previously published results (*Panman et al., 2014*; *Poulin et al., 2014*), *Sox6* territory neurons were confined to the SNc and its subdomains (*Figure 4F*, *Figure 4—figure supplement 1*, see also Supplementary Results). Cells of the *Ebf1* territory were scattered within the periaqueductal grey (PAG), parabrachial pigmented nucleus (PBP), rostral linear nucleus (RLi), and para nigral nucleus (PN; *Figure 4F*, *Figure 4—figure supplement 2*), consistent with previous single-cell studies identifying *Vip*-expressing mDA neurons within the PAG (*Dougalis et al., 2012*; *Hook et al., 2018*; *La Manno et al., 2016*; *Poulin et al., 2014*; *Tiklová et al., 2019*). Cells of the *Pdia5* territory were mapped to the dorsal VTA and the most lateral SN tip (*Figure 4F*, *Figure 4—figure supplement 2*). Consistent with previous analyses (*Hook et al., 2018*; *La Manno et al., 2016*; *Poulin et al., 2014*; *Saunders et al., 2018*; *Tiklová et al., 2019*), *Otx2* territory was localized in the ventral tier of VTA, including PBP, PN, and interfascicular nucleus (IF), but some cells were also found in RLi and caudal linear nucleus (CLi) (*Figure 4F*, *Figure 4—figure supplement 2*). *Gad2* territory was primarily localized in the most medial VTA, especially to IF, RLi, and CLi (*Figure 4A and F*, *Figure 4—figure supplement 3*), supporting findings from previous studies (*Garritsen et al., 2023*; *Poulin et al., 2020*). A few scattered neurons matching the smallest territory, *Fbn2*, were detected in the anteromedial PBP (*Figure 4A and F*, *Figure 4—figure supplement 3*). Finally, neurons of the *Pcsk6* territory were localized broadly in VTA, although being most numerous in PBP (*Figure 4F*, *Figure 4—figure supplement 3*).

The analysis of the anatomical localization of neighborhoods, and how the diversity identified here relates to previous findings, is described in further detail in Supplementary Results related to *Figure 4—figure supplements 1–3*.

## Selective vulnerability to 6-OHDA lesioning

As mentioned, nuclei for snRNAseq were sampled from animals that received a unilateral injection of 6-OHDA in the medial forebrain bundle. To assess the extent of mDA neuron fiber loss in the lesioned hemisphere, striatal tissue was also isolated, sectioned, and analyzed by DAT-binding autoradiography. Animals with partial lesions were chosen for sampling and RNA sequencing of nuclei (Materials and methods, *Figure 5—figure supplement 1*). The nuclei that expressed either *Th* or *Slc6a3* (mDA nuclei dataset, in total 33,052 nuclei) were visualized by UMAP and split by condition to indicate their origin from either lesioned or intact hemispheres (*Figure 5A and B*). Unsurprisingly, fewer nuclei (8243) were recovered from the lesioned compared to the intact hemisphere (24,809). Despite these differences in number of nuclei per animal, the relative UMAP distribution of nuclei appeared similar between individual animals (see https://perlmannlab.org/resources/). Pearson's Chi test of independence further supported this conclusion and showed a similar relative distribution among clusters and neighborhoods, per animal, regardless of the number of mDA neurons remaining (*Figure 5—figure supplement 1C*). Co-staining for TH and RFP on coronal sections of lesioned brains also confirmed this notion (*Figure 5—figure supplement 1E*).

Two conclusions were immediately apparent from the UMAP visualization: First, many nuclei from the lesioned hemisphere were distributed outside of the mDA neuron territories (*Figure 5A–B and F*, see *Figures 2 and 3*), and, as expected, the numbers of nuclei in these territories were less affected by 6-OHDA. Second, most mDA neuron nuclei remaining from the lesioned hemispheres were evenly dispersed together with nuclei from the non-lesioned hemispheres within mDA neuron territories and neighborhoods, indicating that surviving cells were relatively similar to non-lesioned cells at the gene expression level (*Figure 5A and B*). Next, normalized cell loss per territory and neighborhood was calculated based on the sub-clusters cellular composition per condition, accounting for the global FANS yield per condition and sub-cluster size (see Materials and methods, *Supplementary file 5*). The results demonstrated a pervasive loss in the *Sox6* territory (*Figure 5C*), which is consistent with previous studies showing that *Sox6*-expressing cells of the SNc are more vulnerable to pathological

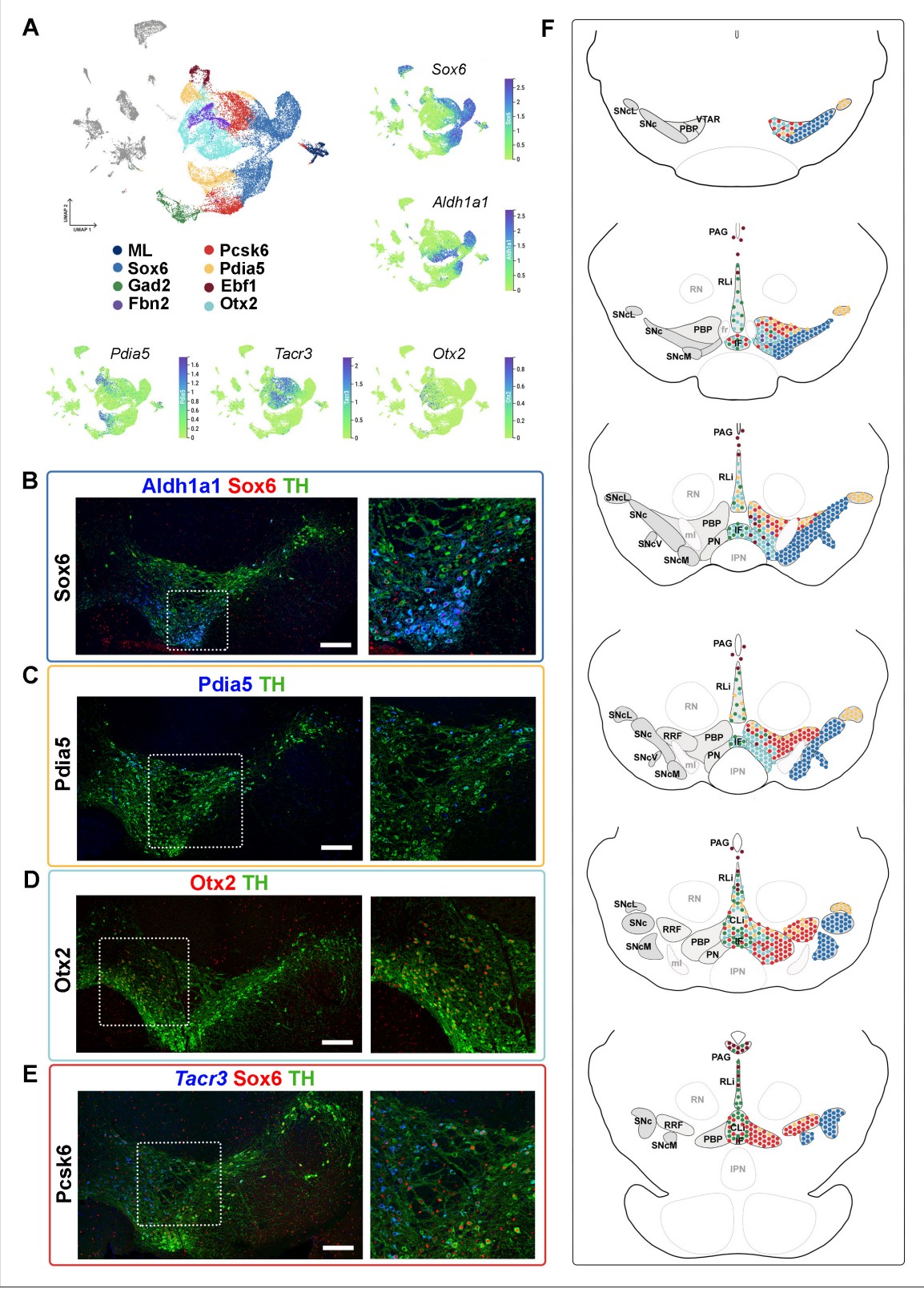

**Figure 4.** Mapping of mDA territories in the adult mouse midbrain. (**A**) A color-coded UMAP projection of the seven mDA territories and of the mostly-lesion (ML) clusters, with individual enriched genes plotted on smaller UMAPs. (**B–D**) Immunohistochemical staining with antibodies indicated in the ventral midbrain. (**E**) Fluorescent RNA in situ hybridization with *Tacr3* probe combined with immunohistochemistry. Insets are shown as higher magnification on the right. (**F**) Schematic presentation of the localization of the seven mDA territories across the midbrain, with the most anterior

*Figure 4 continued on next page*

*Figure 4 continued*

section uppermost. SNc, Substantia Nigra pars compacta, SNcL, Substantia Nigra lateral, SNcV, Substantia Nigra ventral, SNcM, Substantia Nigra medial, VTAR, Ventral Tegmental Area rostral part, PBP, parabrachial pigmented nucleus, PN, paranigral nucleus, PIF, parainterfascicular nucleus, PAG, periaqueductal grey, RRF, retrorubral field, RLi, rostral linear nucleus, CLi, caudal linear nucleus of the raphe, IF, interfascicular nucleus, RN, red nucleus, IPN, interpeduncular nucleus, PN, pontine nucleus, fr, fasciculus retroflexus, ml, medial lemniscus. Scale bars = 200 μm.

The online version of this article includes the following figure supplement(s) for figure 4:

**Figure supplement 1.** Neighborhoods of *Sox6* territory.

**Figure supplement 2.** Neighborhoods of *Ebf1, Pdia5* and *Otx2* territories.

**Figure supplement 3.** Neighborhoods of *Gad2, Fbn2,* and *Pcsk6* territories.

stress in rodents and in PD (*Pereira Luppi et al., 2021*; *Panman et al., 2014*). Cell loss was also notably high in the *Pcsk6* territory. The *Ebf1* territory showed the lowest level of cell loss, while the other territories were affected at intermediate levels (*Figure 5C and D*). Analyses of normalized cell loss were also performed for neighborhoods. Within the *Sox6* territory, a consistent and high level of cell loss was seen in all neighborhoods except *Sox6*_NH3, which showed a somewhat milder but still significant decrease (*Figure 5E*). Notably, different levels of cell loss were often seen within individual territories for example *Gad2*_NH1, and *Otx2*_NH1 as compared to *Gad2*_NH2 and *Otx2*_NH2 (*Figure 5E*). As described above, the corresponding mDA neurons were mostly confined to regions within the VTA, demonstrating that high vulnerability to 6-OHDA was not exclusive to neurons distributed within the SNc.

## Identification of genes associated with vulnerability or resilience

Next, we set out to identify genes that might predict either mDA neuron vulnerability or resilience to 6-OHDA. Because the goal was to identify genes with predictive value expressed in normal mDA neurons, only gene expression in nuclei from the non-lesioned hemisphere was analyzed. Thereby direct effects of the toxin were excluded from the analysis. Accordingly, genes that are commonly and significantly enriched in intact nuclei from clusters with either >90% (vulnerability genes) or <50% (resilience genes) normalized cell loss were identified (*Figure 6—figure supplement 1*, *Supplementary file 5*). Thirty-nine genes fulfilled the stringent cutoff criteria for vulnerability genes. In contrast, only ten genes fulfilled the cutoff criteria for resilience genes (Materials and methods). The selected genes were ranked based on average Log2 fold change, and vulnerability and resilience gene modules were constructed, consisting of 20 and 8 most significantly enriched genes, respectively (*Figure 6F and G*, *Supplementary file 6*). The composite average expression of the gene sets (modules) was visualized in violin plots for neighborhoods and territories (*Figure 6A–D*, *Supplementary file 6*). As expected, the vulnerability and resilience gene module expression in territories and neighborhoods corresponded to the distribution of the clusters used as input for identifying the gene modules (*Figure 6—figure supplement 1*). However, linear regression analysis also demonstrated a highly significant overall correlation between cell loss and the vulnerability module scores (p-value = 3.89e-10), even when the input clusters were excluded from analysis (*Figure 6—figure supplement 1C*). Thus, vulnerability and resilience gene modules can predict cell loss in the data set. This suggests that the identified module genes may be functionally associated with either sensitivity to or protection against 6-OHDA-induced cell stress. One of the vulnerability genes, *Slc6a3*, encodes the dopamine transporter used to import 6-OHDA to the cytosol. Thus, it is unsurprising that neurons of the Gad2 and Ebf1 territories expressing low levels of *Slc6a3* are less vulnerable. However, when regression analysis was repeated using a vulnerability gene module in which *Slc6a3* had been excluded, correlation to cell loss remained highly significant (*Figure 6—figure supplement 1D*, p-value = 1.73e-09). Therefore, the other genes of the vulnerability module predict the sensitivity to toxic stress.

The $Th^+/Slc6a3^+$ dataset of 33,052 nuclei was re-analyzed by the Seurat integration method treating nuclei from lesioned and intact hemispheres as two different experimental conditions (*Stuart et al., 2019*). The main goal of this approach was to identify common transcriptomic states in nuclei from the different conditions via a learned joint structure (see Materials and methods). We used this integrated dataset to validate further the predictive value of the identified vulnerability and resilience gene sets. Vulnerability and resilience modules were plotted across the integrated clusters and were color-coded by territory identities (*Figure 6—figure supplement 2*). In the integrated dataset, the clusters are

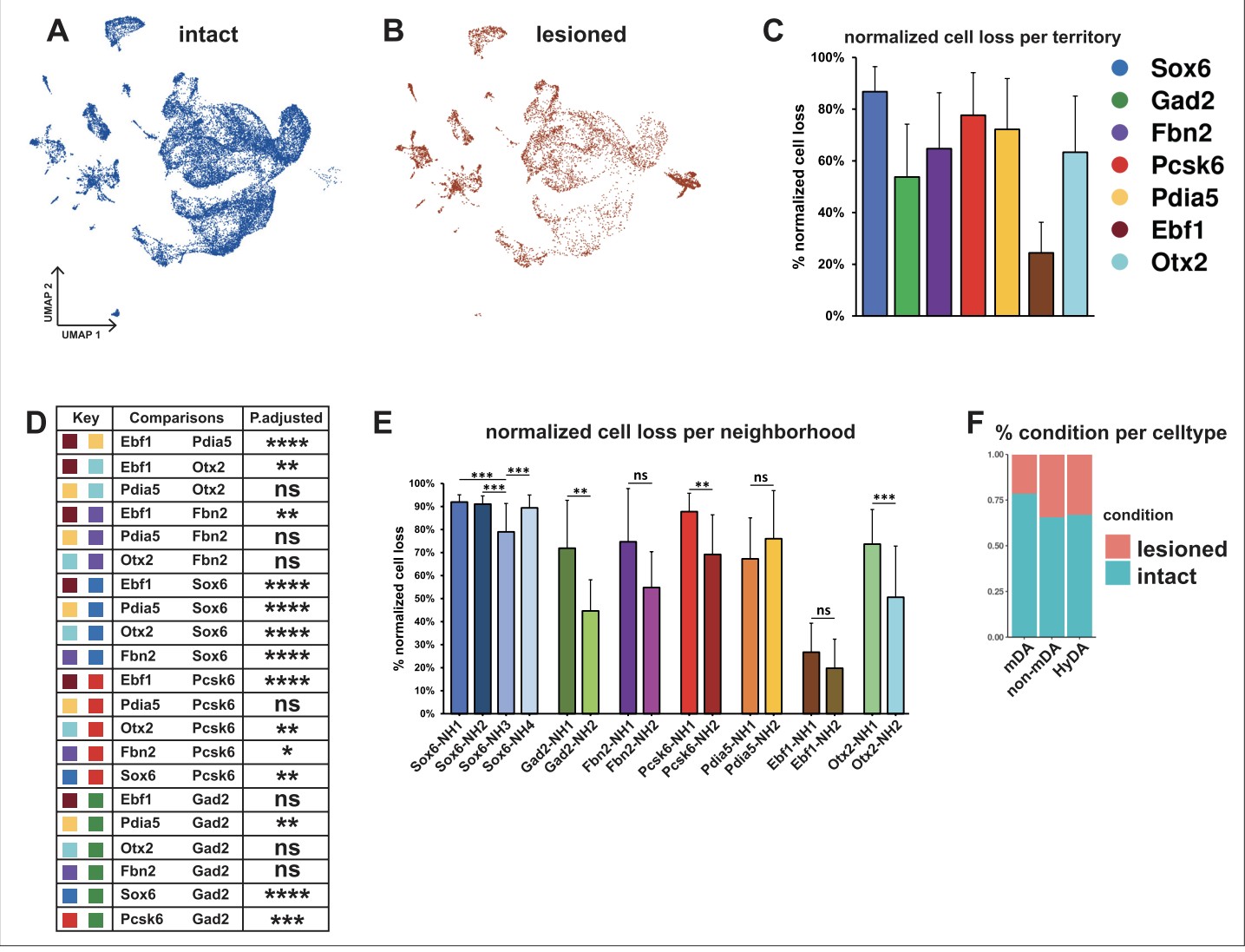

**Figure 5.** Normalized cell loss across territories and neighborhoods in mDA dataset. (**A, B**) UMAP projection of the dopaminergic dataset nuclei from the intact (un-lesioned) and the lesioned hemisphere, respectively. (**C**) Calculated normalized cell loss for territories, visualized as percentages. (**D**) Pairwise comparison of normalized cell loss across territories. (**E**) Calculated normalized cell loss for neighborhoods, visualized as percentages. Comparisons were made between neighborhoods within each territory. See methods for the statistical tests used. (**F**) Percentages of nuclei in each major cell type from either intact or lesioned samples. Sample size per condition (n) = 6, ns = not significant, * p≤0.05, **p≤0.01, *** p≤0.001, **** p≤0.0001. For the Conover-Iman test; p-value = P(T ≥ |t|), and null hypothesis (H₀) was rejected at p ≤ α/2, which is 0.025. Error bars show standard deviation (SD).

The online version of this article includes the following figure supplement(s) for figure 5:

**Figure supplement 1.** Validation of unilateral 6-OHDA-induced lesion.

completely independent of the merged data-based annotations assigned to integrated territories and neighborhoods. Therefore, the distribution of the two modules' scores across the integrated clusters is a good indicator of how predictive they can be, given the territorial identities of the integrated clusters. Of note, the module scores corresponded well to the clusters, as integrated clusters with the *Sox6* territory identity and most of the clusters with the *Pcsk6* territory identity showed considerably high and low values of the vulnerability and resilience modules, respectively.

Conversely, integrated clusters with *Gad2* and *Ebf1* territory identities showed high resilience module values and low vulnerability module values (*Figure 6—figure supplement 2*). In addition, most of the pairwise comparisons of module scores across the mDA-integrated clusters were significant (*Figure 6—figure supplement 2, Supplementary file 7*). Non-mDA clusters were excluded from

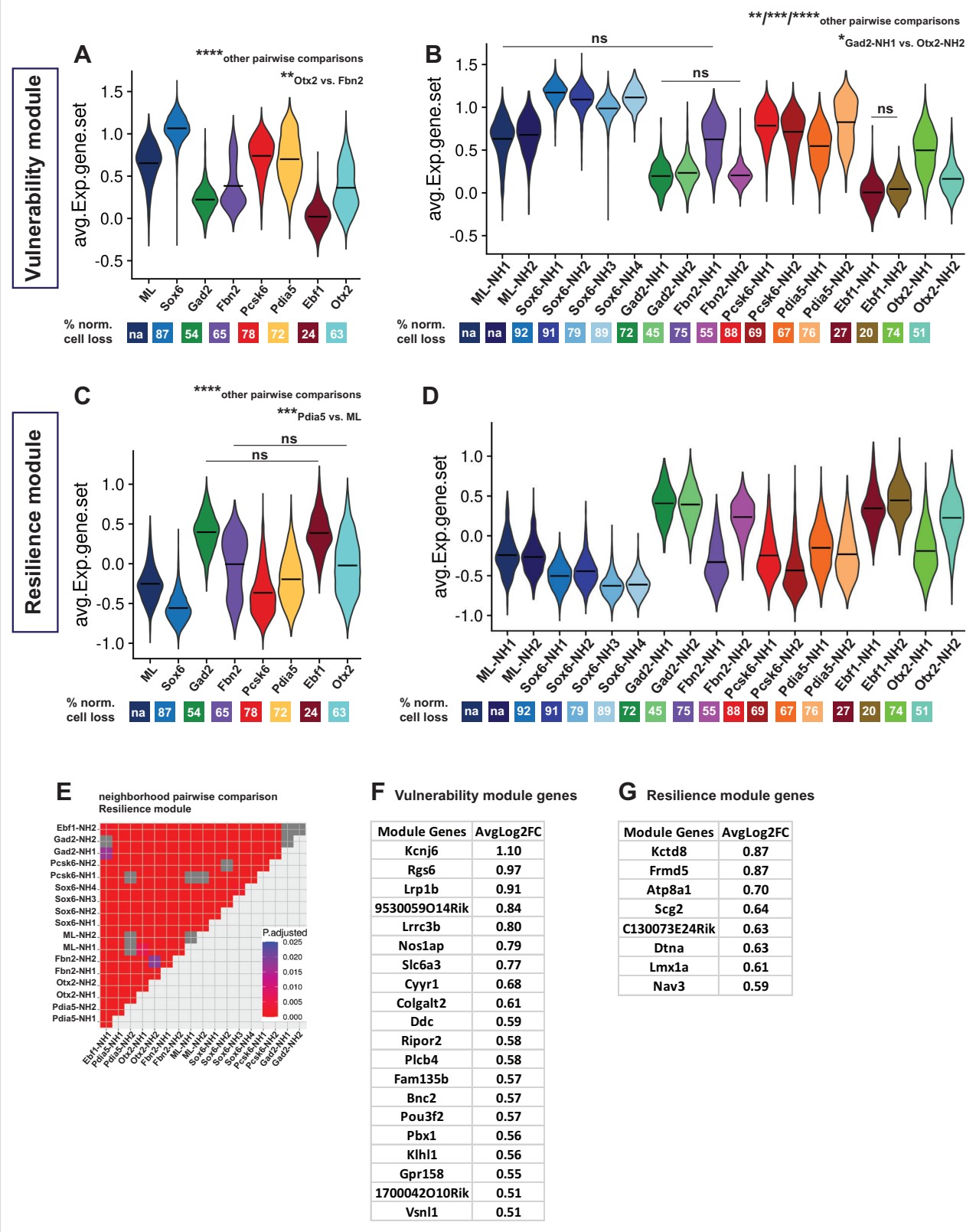

**Figure 6.** Vulnerability and resilience modules in territories, neighborhoods, and the ML clusters. (**A, B**) Violin plots of the vulnerability module across territories and neighborhoods, with percentage of normalized cell loss shown at the bottom per territory and neighborhood, respectively. (**C, D**) Violin plots of the resilience module across territories and neighborhoods, with percentage of normalized cell loss shown at the bottom per territory and neighborhood, respectively. Pairwise comparisons of vulnerability module scores across territories and neighborhoods, respectively. (**E**) Pairwise

*Figure 6 continued on next page*

*Figure 6 continued*

comparisons of resilience module scores across neighborhoods. (**F**) Vulnerability module gene list, ranked by average Log2FC. (**G**) Resilience module gene list, ranked by average Log2FC. Summary statistics center lines are mean values. na = not applicable. ns = not significant. Sample size per condition (n) = 6. See Materials and methods for the statistical tests used.

The online version of this article includes the following figure supplement(s) for figure 6:

**Figure supplement 1.** Normalized cell loss per cluster and its correlation with the vulnerability module.

**Figure supplement 2.** Vulnerability module scores per cluster for integrated *Th⁺/Slc6a3⁺* dataset.

**Figure supplement 3.** Co-expression network analysis of lesioned nuclei from mDA neuron territories.

this statistical analysis. Thus, a strong predictive correlation between the expression of vulnerability and resilience modules is apparent across the snRNA-seq data.

Finally, we used recently developed tools for using WGCNA to calculate gene co-expression modules based on high-dimensional data sets (*Morabito et al., 2023*). Two co-expression modules that shared genes with vulnerability and resilience modules, respectively, were identified. This analysis along with an associated GO term analysis is presented in *Figure 6—figure supplement 3*, and *Supplementary file 9*. Identified GO terms were for both modules associated with neuronal functions such as synaptic functions and ion transport. GO terms associated with glutamatergic and GABAergic neuronal functions characterized the co-expression module mostly similar to the resilience gene module, likely reflecting the resilience of atypical mDA neurons with potential for glutamaterigic and GABAergic neurotransmission.

## Long-lasting gene expression changes in response to 6-OHDA

As mentioned above, cells from lesioned and intact hemispheres were relatively evenly distributed within mDA neuron territories in the UMAP diagram, indicating that most of the surviving cells from the lesioned hemisphere were transcriptionally similar to corresponding cells on the intact hemisphere (*Figure 5A and B*). However, two clusters consisted almost exclusively of nuclei from the lesioned hemisphere. To distinguish them from territories harboring nuclei characteristic of the normal non-lesioned /intact hemisphere of the brain, they were named *mostly lesion* (ML) clusters (*Figure 7A and B*). Although expression of typical mDA neuron markers such as *Th* and *Slc6a3* was low, the ML-clusters were part of the mDA neuron dendrogram branch and shared a parent node with *Sox6*-territory (*Figure 3A and C*). Thus, these nuclei were likely derived from surviving mDA neurons that have become transcriptionally altered due to cellular stress. We took advantage of the integrated dataset described above to analyze this further. Using original territory labels, visualizing the integrated data by UMAP was consistent with data from non-integrated original analysis and demonstrated that the mDA neuron nuclei clustered together (*Figure 7C*, *Figure 6—figure supplement 2*). Next, the integrated data territories were annotated based on the cluster composition of inherited merged (original) territory identities and the enriched markers for the original territories (see *Figure 3C*). Notably, the nuclei forming the ML-clusters were now entirely integrated within the mDA neuron territories, supporting the hypothesis that they originated from healthy mDA neurons. In addition to ML clusters being predominantly made up of lesioned nuclei, *Sox6* expression level and abundance were also clearly higher than *Calb1* (*Figure 7D and E*). Both visualizations of ML-clusters nuclei in the integrated UMAP and quantifications demonstrated that most (>80%) of all ML nuclei were confined to two territories, *Sox6* and *Pcsk6*. It thus seems plausible that the ML nuclei originated from altered neurons in these two territories. Notably, these territories were also the most vulnerable when exposed to 6-OHDA (*Figure 7F–H*, also see *Figure 6A and C*).

Differential gene expression analysis by condition was performed in two different ways. First, gene expression in all nuclei from the lesioned hemisphere was compared to all nuclei from the intact hemisphere. Consistent with the similar distribution of most of the lesioned and intact nuclei in the UMAP, relatively few (242 in total) significantly differentially expressed genes were identified (*Figure 7—figure supplement 1*, *Supplementary file 8*). Second, nuclei from the ML-clusters were compared to nuclei from all other territories. Consistent with a more dramatic and long-lasting influence from lesioning in the ML-clusters, 1577 genes were found to be differentially expressed in this comparison. Several of the most strongly upregulated genes, including *Atf3, Creb5, Xirp2, Clic4, Sprr1a*, and *Mmp12*, have in previous studies been associated with cellular stress, cell death, and cell signaling (*Figure 7I,*

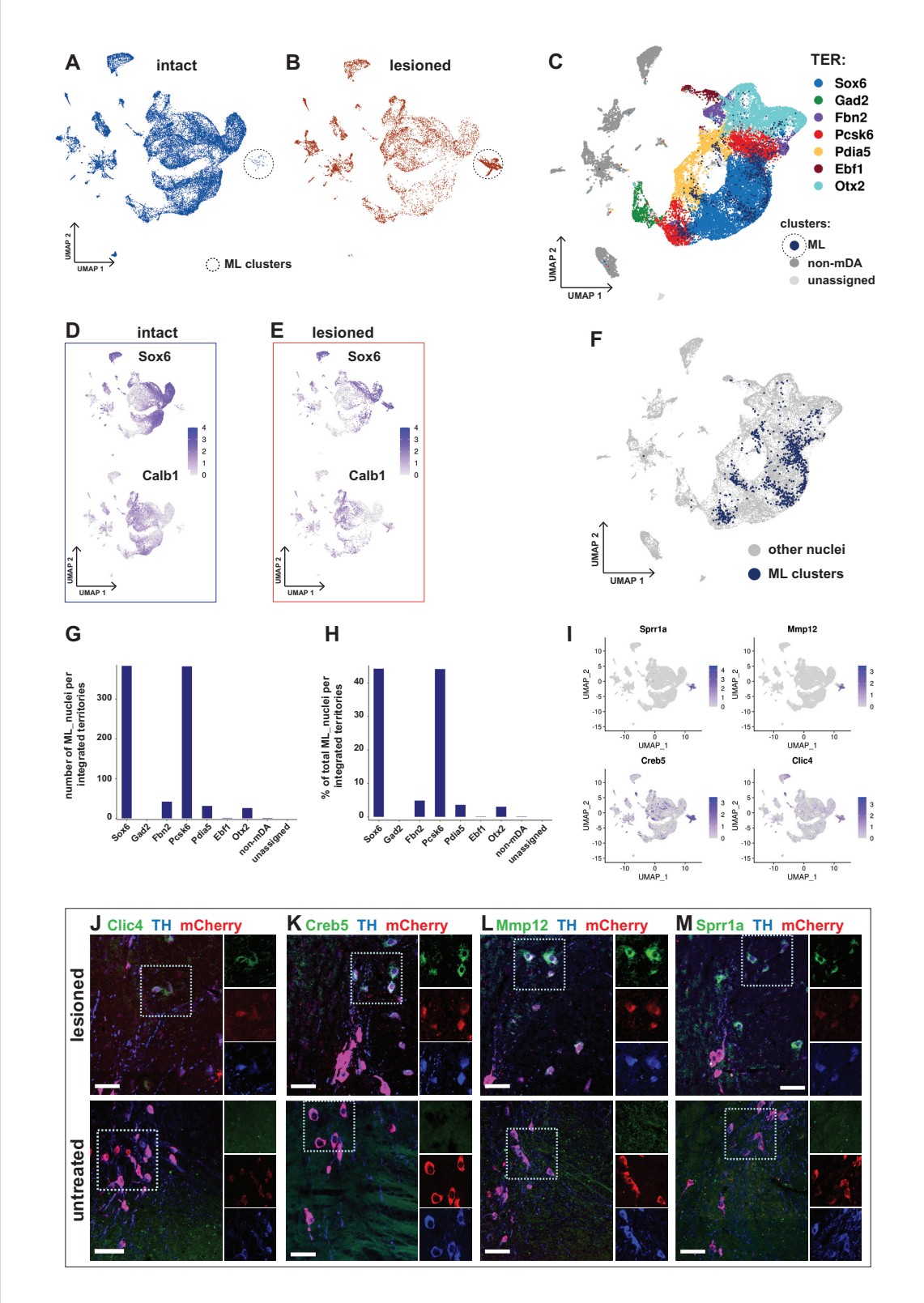

**Figure 7.** Transcriptional kinship and enriched markers of ML clusters. (**A**) UMAP projection of the dopaminergic dataset nuclei from the intact (non-lesioned) hemisphere. (**B**) UMAP projection of the dopaminergic dataset nuclei from the lesioned hemisphere. ML clusters are marked with dashed circles in A and B. (**C**) UMAP projection of the integrated dopaminergic nuclei dataset, color-coded for the territory IDs of the merged dataset. (**D, E**) *Sox6* and *Calb1* expression plotted in intact and lesioned nuclei, respectively. (**F**) Same UMAP projection as in C, with ML clusters nuclei highlighted.

*Figure 7 continued on next page*

*Figure 7 continued*

(**G, H**) Bar graphs showing the numbers and percentages of ML clusters nuclei per territory of the integrated dataset, respectively. More than 80% of ML nuclei cluster in *Sox6* and *Pcsk6* territories of the integrated data. (**I**) Mostly lesioned (ML) clusters enriched markers plotted in UMAP. (**J–M**) ML-enriched markers in I, detected by fluorescent RNA in situ hybridization, combined with TH and mCherry immunohistochemistry in lesioned versus untreated tissue. Scale bar = 50 µm.

The online version of this article includes the following figure supplement(s) for figure 7:

**Figure supplement 1.** ML-clusters enriched genes and GO analysis of ML-specific enriched genes.

**Figure supplement 2.** UMAP, Comparative DE analysis and canonical cell type markers in intact-untreated dataset.

*Figure 7—figure supplement 1*, *Supplementary file 8*). In situ hybridization analysis combined with immunohistochemistry shows apparent upregulation of ML-enriched genes in the lesioned tissue compared to control (*Figure 7J–M*). A GO-enrichment analysis of the genes uniquely dysregulated in ML clusters identified genes associated with axon guidance, synaptic transmission and RAS signaling pathway, among other terms (*Figure 7—figure supplement 1D, E*, *Supplementary file 10*).

Although 6-OHDA was injected unilaterally, a potential influence on cells in the intact hemisphere could interfere with some of our conclusions. Therefore, to assess whether the stereotactic injection had any effect on the transcriptional profile of the nuclei from the intact hemisphere, we created and analyzed a separate dataset including all $Th^+$/ $Slc6a3^+$ nuclei from the intact hemisphere of the brain, merged with a new dataset of 6001 mCherry$^+$ nuclei isolated from the ventral midbrain of untreated mice. UMAP visualization of this combined data set indicated an almost complete overlap between the two groups of nuclei. The dataset was annotated similar to the procedure above and all major cell types were also identifiable in it (*Figure 7—figure supplement 2*). Moreover, comparative differential gene expression analysis also supported the conclusion that any differences between the two groups were attributable to undefined technical parameters in experimental conditions rather than to the effects of the toxin (*Figure 7—figure supplement 2*, *Supplementary file 8*).

## Relationship between dopamine neuron groups in the mouse and human brain

Understanding how the emerging knowledge of mouse mDA neuron diversity compares to the molecular landscape in the human brain is of major interest. We, therefore, integrated and reanalyzed our mouse data with human snRNAseq data from a recently published paper (*Kamath et al., 2022*). Canonical correlation analysis (CCA) was used to integrate 25,003 nuclei from human control and diseased (PD and Lewy body disease) donors with our lesioned and intact nuclei from the $Th^+$/ $Slc6a3^+$ dataset (see Materials and methods). UMAP visualization revealed a relatively large overlap between nuclei from the two species, despite some regions being predominated by one specific species or condition. Detection of mouse-specific UMAP regions is not surprising, given the comprehensiveness of the mDA-enriched mouse dataset. However, human-specific UMAP regions were also observed (*Figure 8*). Next, we used the mouse intact nuclei from the mDA territories (see *Figure 3*) to build a reference UMAP structure onto which the human control nuclei (query dataset) were projected (see Materials and methods). The results demonstrated that almost 75% of the human control nuclei are mapped to mouse *Sox6* territory (*Figure 8—figure supplement 1A–D*). Upon using the mouse intact nuclei, only from *Sox6* territory as a reference, and the human control nuclei that were mapped to *Sox6* territory in the previous step as query, most of this human query subset were mapped to *Sox6*-NH2, while almost no nuclei mapped to *Sox6*-NH4 (*Figure 8—figure supplement 1E–H*). The findings that most human nuclei bear the closest resemblance to mouse *Sox6* territory nuclei were expected, as the human nuclei were derived from SNc.

## Discussion

When histochemical methods for mapping catecholaminergic cells and projections were developed (*Falck et al., 1962*), it became clear that mDA neurons consist of heterogeneous groups of cells with distinct anatomical locations and innervation targets. However, we now know that a substantial diversity exists beyond the anatomically defined cell groups, supported by studies focused on mDA neurons using rapidly emerging single-cell methods to characterize cellular diversity (*Garritsen et al.,*

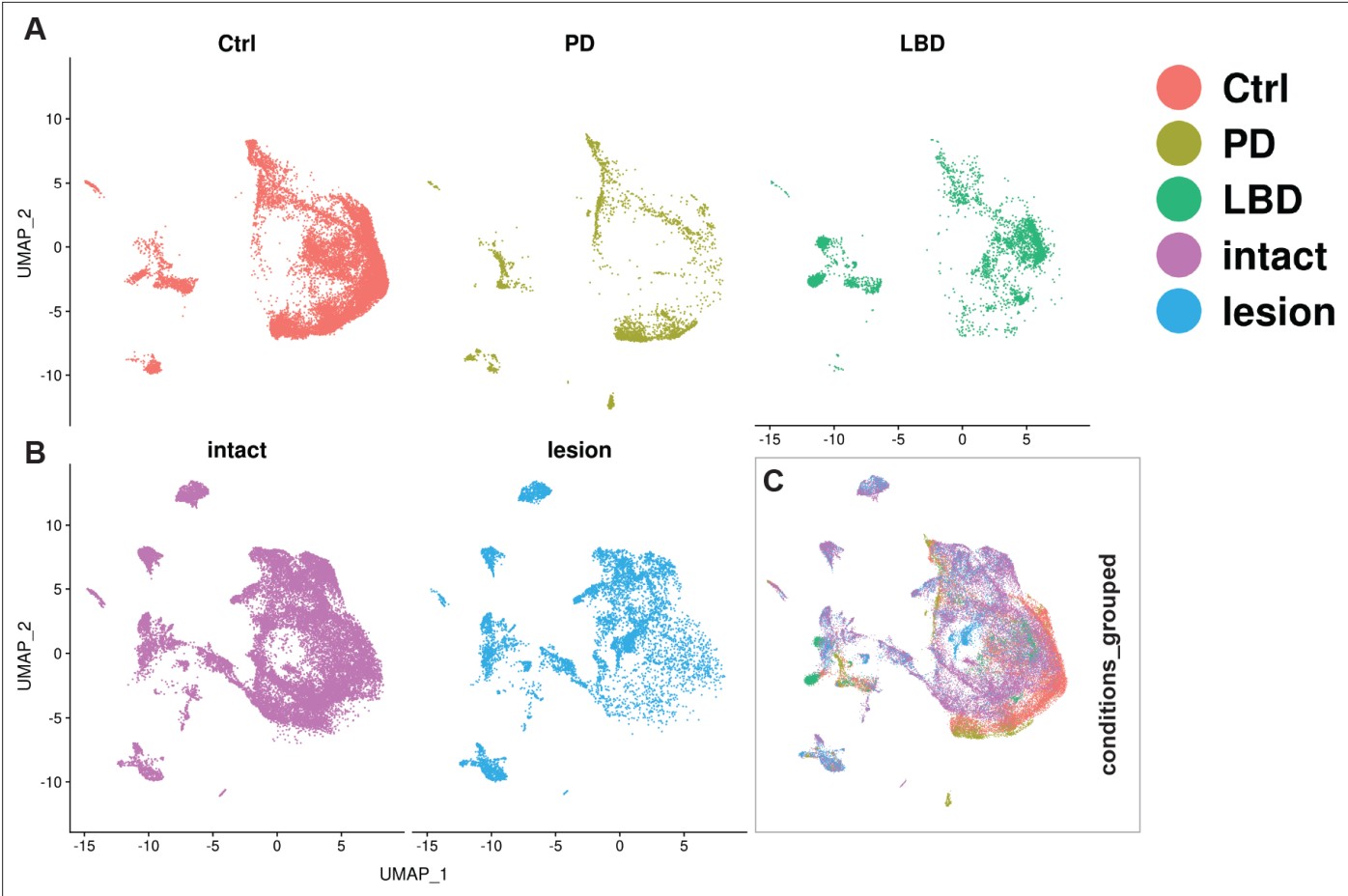

**Figure 8.** Integration of human and mouse datasets. (**A, B**) UMAP projections of the integrated dopaminergic human (control +diseased) and mouse (intact +lesioned) nuclei, split per condition. This integrated dataset includes 25,003 human nuclei and 33,052 mouse nuclei. (**C**) UMAP projections of the human and mouse nuclei grouped by condition. Dataset was generated using the Seurat canonical correlation analysis (CCA). Ctrl = control neurotypical, PD = Parkinson's disease, LBD = Lewy body dementia, intact = non-lesioned.

The online version of this article includes the following figure supplement(s) for figure 8:

**Figure supplement 1.** Integrated mouse-human mDA dataset.

*2023*; *Hook et al., 2018*; *La Manno et al., 2016*; *Poulin et al., 2020*; *Poulin et al., 2014*; *Tiklová et al., 2019*). Here, the goal was to create a comprehensive gene expression atlas over the mouse mDA neuron gene expression landscape by using snRNA-seq for the analyses. In addition, we wished to address the differential vulnerability to neuronal stress within the diverse groups of mDA neurons.

Unsurprisingly, global gene expression patterns differ substantially between cell types such as astrocytes, neurons, or endothelial cells. In addition, different neuron types with distinct developmental origins usually differ substantially in scRNA sequencing analyses. Thus, DA neurons from the hypothalamus, which have a distinct developmental origin from mDA neurons, differ from mDA neurons by many differentially expressed genes. Our study was primarily motivated by the more challenging need to describe differences across the population of related mDA neurons, which have a common developmental origin and a high degree of molecular similarities. The challenge in the analysis is evident from UMAP visualizations, as presented here, where mDA neurons are distributed in a shifting gene expression landscape without clearly distinct borders between groups of mDA neurons. Thus, defining distinctive sub-types within such a nuanced gene expression landscape may be misleading, as noted in several previous single-cell studies (*Kozareva et al., 2021*; *Saunders et al., 2018*; *Tasic et al., 2018*; *Yao et al., 2021*). Therefore, we chose a nomenclature that better reflects the continuity between mDA neuron groups by referring to categories of mDA neurons as belonging to 'territories' and 'neighborhoods'. We do not propose that the taxonomy presented here should be viewed as

universal or a definitive endpoint, but rather should be seen as high-resolution gene expression analysis that will help to guide further studies that use strategies to understand connectivity, neurophysiology, and behavior in investigations of relevant neuronal sub-circuits influenced by mDA neurons.

The mDA neuron atlas presented here is based on the sequencing of a large number of mDA neurons (40,000 in total), which was achieved by using snRNA-seq, which is superior to scRNA-seq for reducing sampling bias. Furthermore, in support of a robust sampling, a large proportion of mDA neurons population was analyzed from each dissected brain (consistently above 20%), a 50–100 times improvement from our previous single-cell study (*Tiklová et al., 2019*). Indeed, the neurons belonging to the SNc (*Sox6* territory) corresponded to roughly one-third of all mDA neurons, a proportion that conforms to previous stereological cell counting of mouse mDA neurons (*Björklund and Dunnett, 2007*; *Zaborszky and Vadasz, 2001*). Thus, we assume that the cellular stoichiometry, as presented in the present snRNA-seq atlas, roughly corresponds to the distribution of mDA neurons in the adult mouse brain.

Previous scRNA-seq studies have described mDA neuron groups characterized by gene expression patterns that resemble those in different territories and neighborhoods. Comparisons to those studies are described and referenced in Supplementary Results and show that distinguishing gene expression features seen in six of the seven territories described here are similar to those found in previous single-cell studies (*Hook et al., 2018*; *La Manno et al., 2016*; *Poulin et al., 2014*; *Saunders et al., 2018*; *Tiklová et al., 2019*). However, the presented mDA neuron atlas illustrates previously unknown diversity within these groups. Thus, the *Sox6* territory was divided into four neighborhoods with partly unique anatomical localizations within the SNc. In addition, the gene expression signature of the *Pcsk6* territory, comprising a relatively large number of cells, did not correspond well to previously described mDA neuron groups. Thus, the *Pcsk6* territory neurons apparently correspond to an mDA neuron group previously not recognized.

Moreover, while *Vip*⁺ mDA neurons were identified in several previous single-cell studies, we show here that the *Ebf1* territory can be divided into two related neighborhoods encompassing *Vip*⁺ and *Col24a1*⁺mDA neurons, respectively. Although these cells were most enriched in PAG, they were scattered also in other regions such as PBP and CLi. *Col24a1*⁺likely corresponds to a specific population of *Npw*⁺ mDA neurons, localized in PAG, VTA, and the dorsal raphe, which project to the extended central amygdala and modulate mouse behavior under stress (*Motoike et al., 2016*).

We also extended our analysis to neurons exposed to neuronal stress by sampling nuclei from the lesioned after partial unilateral exposure to the neurotoxin 6-OHDA. The mechanisms of neuronal stress leading to the degeneration of mDA neurons in PD differ from events following exposure to neurotoxins (*Blesa and Przedborski, 2014*). However, it is known that most insults to mDA neurons, including those resulting from 6-OHDA exposure, affect SNc more severely than neurons of the VTA, indicating an inherent higher vulnerability of these neurons. This is evident in our analysis, where normalized cell loss was most severe within the *Sox6* territory. However, a notable high vulnerability was also seen outside the *Sox6* territory, for example within the *Pdia5* and *Pcsk6* territories. Indeed, the *Pcsk6*_NH1, apparently localized within the PBP, is even more vulnerable than the *Sox6*_NH3. Within the VTA territories, resilient neighborhoods included several that express Slc17a6 encoding the glutamate transporter VGLUT2, a protein linked to increased and decreased dopamine neuron vulnerability (*Kouwenhoven et al., 2020*; *Steinkellner et al., 2018*).

The information on normalized cell death was used to identify genes enriched in highly vulnerable or resilient clusters. Regression analysis showed a highly significant correlation between the vulnerability module scores and normalized cell loss in neuron groups, that is clusters that were not used as input to identify the genes used in calculating the module. We note that several genes in these modules may be functionally linked to either vulnerability or resilience: *Kcnj6* is a known marker for SNc neurons and has, in previous studies, been shown to be associated with the vulnerability of these neurons (*Liss et al., 2005*). *Rgs6*, regulator of G-protein signaling 6, has been implicated in age-dependent α-synuclein accumulation and mDA neuron degeneration when mutated in mice (*Luo et al., 2019*). *Nos1ap*, nitric oxide synthase 1 adaptor protein, may play a role in excitotoxic neuronal damage in neurodegenerative disease (*Wang et al., 2016*). *Atp8a1*, ATPase phospholipid transporting 8A1, may be implicated in neuroprotective effects caused by pumping phosphatidylserine across the plasma membrane (*Wang et al., 2023*). In addition, the protein encoded by the *Vsnl1* gene is a calcium sensor that is known to be increased in cerebrospinal fluid in Alzheimer's patients and

may be linked to disease progression (*Liao et al., 2022*; *Zetterberg, 2017*). Therefore, the analysis of 6-OHDA vulnerability complements the atlas in ways that will be valuable for understanding how the molecular gene expression landscape influences sensitivity to pathological stress. It will be important in future studies to extend these studies to mouse and human neurons exposed to pathology that more resembles that seen in PD.

To avoid extensive influence on gene expression from the acute toxin effects in dying cells, brain tissue was collected for nuclei isolation several weeks after 6-OHDA exposure. Most nuclei derived from the lesioned hemispheres of dissected brains were scattered in UMAP analysis with nuclei from the intact hemisphere, suggesting that the toxin had not caused any severe gene expression alterations. However, nuclei in the ML clusters were an exception, as they were derived almost exclusively from neurons dissected from the lesioned hemisphere. These toxin-exposed neurons displayed more profound and long-lasting alterations in gene expression. We addressed the origin of these cells in an integrated data set, and it is plausible from this analysis that most of these altered neurons were derived from neurons belonging to the two most vulnerable *Sox6* and *Pcsk6* territories. Gene expression changes in ML-clusters' included severe down-regulation of typical mDA neuron gene markers such as *Th* and *Slc6a3*. In addition, many enriched genes have previously been associated with neurodegeneration or response to neuron stress. For example, *Atf3, Lifr, Sprr1a* have been reported to be part of a regeneration-associated gene program in peripheral nerves (*Kanaan et al., 2015*; *Seijffers et al., 2007*). *Atf3* is also known to be upregulated as a result of activation of the unfolded protein response pathway via PERK (*Jiang et al., 2004*). *Ngf*, encoding the nerve growth factor, is known to be neurotrophic to mDA neurons and is dysregulated in PD (*Mogi et al., 1999*). Thus, part of this stress response is likely to be neuroprotective and likely reflects a cellular survival response in affected cells.

In conclusion, the work presented here provides a comprehensive atlas over mDA neuron diversity in mice. Combined with the included anatomical analysis and assessment of neuron vulnerability after toxin stress, we anticipate that this will be a useful resource in studies to understand the intricate functions of neurocircuitries involving dopamine neurotransmission functionally.

## Materials and methods
### Mouse lines and genotyping
*Slc6a3^Cre* (*Ekstrand et al., 2007*), *Pitx3^GFP* (*Zhao et al., 2004*) *Rpl10a-mCherry* (*TrapC^fl*) (*Hupe et al., 2014*) and their genotyping have been described before. *Slc6a3^Cre* and *TrapC^fl* lines were intercrossed to generate *Slc6a3^Cre/+; TrapC^fl/fl* mice. Mice were injected at the age of 4.5–5 months. Untreated (WT) mice were 3 months (n=3) and 18 months (n=3) old. All mice were female. All experimental procedures followed the guidelines and recommendations of Swedish animal protection legislation and were approved by Stockholm North Animal Ethics board (animal permit number: 13830/18).

### Partial lesions with 6-OHDA
C57BL6 female mice were pretreated (30 min before surgery) with 25 mg/kg desipramine (i.p.; Sigma-Aldrich) and 5 mg/kg pargyline (i.p.; Sigma-Aldrich), anesthetized with 2–2.5% isoflurane, placed in a stereotaxic frame, and unilaterally injected, over 2 min, with 0.7 or 1.5 µg of 6-OHDA in 0.01% ascorbate (Sigma-Aldrich) into the medial forebrain bundle (MFB) of the right hemisphere. The coordinates for injection were AP, −1.1 mm; ML, −1.1 mm; and DV, −4.75 mm relative to bregma and the dural surface (*Paxinos and Franklin, 2019*). The tissues were collected 6 weeks after 6-OHDA administration.

### DAT binding autoradiography
Fresh frozen thaw-mounted striatal sections (12 µm thick) were initially incubated in 50 mM Tris–HCl/120 mM NaCl (pH 7.5) for 20 min. Then the same sections were incubated in binding buffer (50 mM Tris– HCl/120 mM NaCl, pH 7.5/1 µM fluoxetine) containing 50 pM of the DAT ligand ([125]I) RTI55 (PerkinElmer Life Sciences, Boston, USA) for 60 min. To detect nonspecific binding, 100 µM nomifensine was added to the assay. The sections were washed 2 × 10 s in ice-cold binding buffer, rapidly dipped in deionized water, air-dried, and exposed to autoradiographic films in X-ray cassettes at room temperature for 2 days.

## Fluorescent RNA in situ hybridization and immunohistochemistry

Mice were deeply anesthetized, and the brains collected after intracardial perfusion with +37 C PBS, followed by 4% PFA in PBS. The midbrain piece was coronally cut out using a brain matrix and post-fixed for 2–3 days in PFA at RT, changing the PFA solution daily. The tissues were dehydrated and processed into Histosec wax (Merck) using an automated tissue processor (Leica TP1020), and coronal 6 µm sections were collected at Superfrost Plus slides (Thermo-Scientific) using an automated water-fall microtome (Epredia HM355S).

Immunohistochemistry and fluorescent RNA in situ hybridization were performed essentially as described previously (*Tiklová et al., 2019*), except for Pdia5 antibody Triton X-100 in the washing buffers and blocking solutions was replaced by 0.1% Tween-20. For the analysis of *Tacr3*, parallel slides were stained with an antibody for Aldh1a1. Two to three animals were used for each experiment. Each ML-territory probe was tested on lesioned (n=3) and control *Slc6a3^Cre; TrapC ^fl/fl* mice.

The ML-clusters specific probes *Clic4, Creb5, Mmp12,* and *Sprr1a* were cloned fragments from the respective protein coding sequences of mouse cDNA. Size of fragments: *Clic4* 722 bp, *Creb5* 591 bp, *Mmp12* 759 bp and *Sprr1a* 598 bp. The probe used for *Vip* was cloned with primers from Allen Mouse Brain Atlas (experiment 77371835). The *Gad2* probe was a cloned 869 bp fragment from the protein coding sequence of mouse *Gad2* cDNA. *Tacr3* probe was a 992 bp Genestrings fragment (Invitrogen) from the protein coding sequence of mouse *Tacr3* cDNA. All cDNA fragments were cloned into pCR-Blunt II TOPO vector (Invitrogen 450245) and DIG-UTP labelled cRNA probes synthetized from linearized plasmids using either SP6 or T7 RNA polymerases (Roche) according to manufacturer's instructions. The list of Allen Mouse Brain Atlas experiments of which images were shown in *Figure 4* and in *Figure 1—figure supplement 3*, *Figure 4—figure supplements 1–3*, as well as the list of primary and secondary antibodies used, are in Supplementary Materials (*Supplementary file 3*).

## Image capture and processing

Images of fluorescent in situ hybridizations and immunostainings were taken using a Zeiss confocal microscope LSM700. Tiling of the individual images was done with Fiji (ImageJ) Pairwise stitching plug-in *Preibisch et al., 2009*. Brightness and contrast were adjusted after tiling, if needed.

## In situ RNA sequencing

The in situ RNA sequencing experiments were performed as described (*Gyllborg et al., 2020*). In brief, fresh-frozen 10 µm thick mouse brain sections were sectioned on a cryostat and stored at –80 °C until fixation. After fixation in 3% (w/v) paraformaldehyde (Sigma) in DEPC-treated PBS (DEPC-PBS) for 10 min at RT, tissues were washed in DEPC-PBS and reverse transcription was performed at 37 °C for 16 hr. Tissues were then treated with RNaseH at +37 °C for 45 min to make single-stranded cDNAs, followed by padlock probe (IDT Corelville, Iowa; 4 nmole, standard desalting, 5 prime phosphorylated; *Supplementary file 4*) hybridization and ligation at 45 °C for 60 min. Rolling circle amplification was performed at 30 °C for 16 hr. Rolling circle products were detected by using fluorescently labeled detection oligo (*Supplementary file 4*, IDT Corelville, Iowa).

## Imaging and data processing of ISS

Images were acquired with Leica DMi8 epifluorescence microscope equipped with an external LED light source (Lumencor SPECTRA X light engine), automatic multi-slide stage (LMT200-HS), sCMOS camera (Leica DFC9000 GTC), and objective (HCX PL APO 40 X/1.10 W CORR). A series of images (10% overlap between neighboring images) were obtained, and maximum intensity projected to two-dimensional images. These images were aligned between cycles and stitched together using Ashlar algorithm. Stitching was followed by retiling to get smaller (6000x6000 pixel) images. Those images were used for decoding using PerRoundMaximumChannel Decode Spots Algorithm. The resulting spots were filtered based on minimum quality. The preprocessing (*Langseth et al., 2021*) and decoding (*Langseth and Marco, 2021*) pipeline can be found on the Moldia GitHub page (https://github.com/Moldia).

## Tissue processing for FANS

*Slc6a3^Cre TrapC ^fl/fl* (n=6 for lesion cohort, n=6 for control cohort) mice were euthanized with $CO_2$, and brains were rapidly removed and transferred into cold PBS. The midbrain was dissected and

snap-frozen on dry ice. For lesioned brains, the lesioned hemisphere was separated from the intact hemisphere before being snap-frozen. Tissue processing for FANS was performed as described previously (*Toskas et al., 2022*), with the addition of adding 0.1% Triton and 5 mM $CaCl_2$ to the lysis buffer. For the control cohort, resuspended and filtered nuclei were stained with NeuN-647 conjugated (1:1000, MAB377 Millipore Corp) antibody for a few hours before being sorted. Nuclei were sorted using a FACSAria Fusion or FACSAriaIII cell sorter with the FACSDiva software (BD Biosciences). The nuclei were identified by forward- and side-scatter gating, a 405 nm laser with a 450/50 or 450/40 filter, a 561 nm laser with a 610/20 filter, and a 633 nm laser with a 660/20 filter, quantifying DNA content per event to assure that only singlets were collected. A 100 µm nozzle and an acquisition rate of ~3000 events per second was used. Nuclei were stained with DAPI (1:200, Merck) 5 min before sorting and preserved in 2% BSA until downstream procedures.

## Library preparations and sequencing

Ventral midbrains tissues from 6 mice (6 lesioned, 6 non-lesioned (intact) hemispheres) were used to obtain 70,464 nuclei (36789 un-lesioned [intact] and 33,675 lesioned). Ventral midbrain tissue from 6 control mice were used to obtain 8311 nuclei. Chromium Single Cell 3′ v3, dual indexing (10 x Genomics) was used to make single-nucleus libraries, in accordance with the manufacturer's protocols. Libraries were sequenced on a NovaSeq 6000 system (ESCG, SciLifeLab, Karolinska Institute). Library preparations and sequencing for the nuclei from the intact and lesioned ventral midbrain hemispheres were done together but separately from the control nuclei. snRNAseq analysis.

Sequenced reads were demultiplexed and aligned to a CellRanger customized reference package 'transcriptome: mm10-3.0.0_premrna', with pre-mRNA added to ensure both premature and mature mRNAs are mapped accurately, using CellRanger Pipeline version 3.1.0 from 10 x Genomics (*Zheng et al., 2017*). A count matrix (filtered_feature_bc_matrix) per sample was generated using CellRanger Pipeline version 3.1.0. Seurat package version 4 was used for downstream analyses of the data (*Hao et al., 2021*). Seurat Read10x() was used to load the sparse count matrices of 10 X genomics Cell-Ranger output and CreateSeuratObject() function was used to make one Seurat Object per sample. All 12 samples (6 lesioned, 6 un-lesioned) were merged to generate one Seurat object for the entire data. Quality control and filtering of data were done in the following steps. First, the percentage of transcripts that map to mitochondrial genes and ribosomal genes were calculated using Seurat function PercentageFeatureSet and the corresponding values added to the created percent_mito% and percent_ribo% metadata columns, respectively. Then, CellCycleScoring function was used to score cell cycle phases, and phase metadata was calculated and added based on the expression of G2-M and S phase canonical markers. In the next step, the ratio of a gene's unique molecular identifier (UMI) counts to the sum of all UMIs per library (nucleus), was calculated as the average relative expression of each gene per library. *Malat1*, a lncRNA gene, was expressed in all libraries with a mean fraction of total UMI counts per library of about 7%. Polyadenylate captured RNA-seq libraries, irrespective of the used protocol, show a relatively high detection of *Malat1* reads. However, compared to other methods, the detection is even higher in snRNA-seq due to its nuclear localization and therefore *Malat1* expression is used to estimate the nuclear proportion of total mRNA (*Bakken et al., 2018*). *Malat1* was filtered out for downstream analysis. Genes with reads present in fewer than 10 nuclei (libraries) were removed. Libraries with detected genes below 500 and above 10,000 (doublets), and with percent_mito above 5% were also removed from the analysis.

## All-nuclei dataset

After the filtering steps (above), 68,914 nuclei remained (36,051 intact, 32,863 lesioned). The remaining nuclei after the quality control constituted the all-nuclei dataset and had UMIs per library in the range of 574–122,753, genes per library in the range of 501–9991, with average UMIs per library of 17,630, and average genes per library of 4762. The average UMI/gene was 3.58, with a range of 1.14–13.31 (*Figure 1—figure supplement 2*).

## Dopaminergic nuclei dataset

The filtered dataset (all-nuclei, above) was then subject to selection based on the expression of two genes, *Th* and *Slc6a3* (DAT). Nuclei expressing either *Th* or DAT (*Th+/ Slc6a3+*) were selected. The

remaining 24809 and 8243 nuclei from the respective intact and lesioned hemispheres were used to create the dopaminergic dataset.

## Combined intact and control nuclei dataset

Count matrices for the 6 intact samples as well as the 6 control samples were loaded separately and merged to create a dataset of 12 samples (6 intact, 6 control). After the QC steps, described above, and selection of libraries based on either *Th* or DAT *(Th⁺ | Slc6a3⁺)* expression, 24,809 nuclei from the intact hemisphere and 6001 nuclei from the control remained which were used to make the final intact-control dataset. Interestingly, the ratio of the *Th⁺ | Slc6a3⁺* filtering for both groups was 0.7.

## Normalization, variance stabilization, and dimensionality reduction

SCTransform package (*Hafemeister and Satija, 2019*) was used to normalize, scale, and find the top 1000 variable genes, with the intension to replace the heuristic approaches of adding pseudo-counts and log transformation. This method of normalization and variance stabilization is meant to improve the disentanglement of biological heterogeneity from the confounding effects of the technical variables, using the Pearson residuals of the regularized negative binomial regression, with the sequencing depth as a covariate in a generalized linear model. The residuals of the SCTransform were used as input for dimensionality reduction. Principal component analysis (PCA) was applied, followed by the JackStraw procedure and ElbowPLot to visualize the standard deviations of the principal components. PCs 1:30 were used in the UMAP embedding.

## SNN graph-based Clustering

Seurat FindNeighbors function was applied for SNN graph construction, followed by FindClusters function, using the Louvain algorithm, to identify the clusters. For the all-nuclei dataset, the resolution 0.1 was used since the goal was to identify the major cell classes. BuildClusterTree function was used to make a phylogenetic tree, with assay = "SCT", slot = "data" parameters for the all-nuclei dataset. For the different datasets in this research, a range of resolutions were tested to estimate the optimal cluster numbers, as a heuristic approach based on known markers, neuroanatomical regions and hierarchical dendrograms. These estimated values were used in determining k (centers) parameters in k-means clustering when applicable (below).

## K-means and hierarchical clustering

A matrix of PCA values based on PCs 1:30 was generated to be used as input for the kmeans clustering function, with 'MacQueen' algorithm. The computed clusters (centroids) were added to the object metadata and then were subject to hierarchical clustering, with distance matrix computation method set as 'Euclidean' and the agglomeration method as 'ward.D2'.

## Integration analysis mouse dopaminergic dataset

For the dopaminergic dataset, it was split based on condition (intact / lesioned), SCTransform approach was used to independently, normalize and identify the top 1000 variable genes per condition. Same number of genes (1000) were selected as integration features to be used as 'anchors' in data integration, using SCT as the normalization method. PCs 1:30 were used in the UMAP embedding. K-means and hierarchical clustering were performed as described above.

## Calculation of cell loss

The ratio of the FANS yield for the nuclei passing the QC, for condition (intact to lesioned) was calculated and the quotient (1.097) was set as a yield normalization coefficient ($\beta$=1.097) for lesioned subclusters. The calculation of cell loss was done per subcluster for the original kmeans clusters in the dopaminergic nuclei dataset. Based on the hierarchical dendrogram and class annotation, the non-dopaminergic clusters were excluded from the analysis. Louvain modularity optimization algorithm was applied with resolution of 0.5 to obtain the subclusters. Most clusters were resolved into subclusters with no or only one identified singleton, while no clusters were resolved into subclusters with higher than 3 singletons. The cell composition of each subcluster per condition was determined, considering the FANS yield coefficient. For each cluster, the average normalized cell loss was calculated, considering the subcluster size, according to the formula below, with normalized cell loss per

cluster; A, determined by subcluster size by condition; $s_l$ and $s_i$ lesioned and intact respectively, N; number of subclusters per cluster, and β.

$$A = (\sum_{s=1}^{s=n} 1 - ((\beta \times sl) \div si)) \div N$$

The A=1 denotes total cell loss while A=0, indicates no cell loss at all. The subcluster cell loss values were used to calculate the normalized cell loss per cluster, neighborhood, and territory. Subclusters with $1 - \frac{(\beta * sl)}{si} < 0$, were set to NA. The statistical tests used for cell loss analyses include Shapiro-Wilk normality test, either Levene's Test or Fligner-Killeen (for non-normally distributed data) tests of homogeneity of variances, Kruskal-Wallis rank sum test, and the Conover-Iman test, with Benjamini-Hochberg p adjustment method, for multiple pairwise comparisons across clusters, territories, and the neighborhoods within *Sox6* territory. For all other territories of only two neighborhoods, Levene's Test for Homogeneity of Variance and either Two Sample t-test or Wilcoxon rank sum test were applied. For the Conover-Iman test, the default (altp = F ALSE) was set as p-value = $P(T \geq |t|)$, and the null hypothesis ($H_0$) was rejected at $p \leq \alpha/2$, which is 0.025.

## Vulnerability and resilience modules

Vulnerable (normalized cell loss >90%) and resilient (normalized cell loss <50%) clusters were identified and gene sets for each module were created based on being commonly upregulated (adjusted p-value < 0.05 and average Log2FC > 0.5) in the intact-only nuclei of either the vulnerable or resilient clusters. The Seurat function AddModuleScore was used to calculate the module scores. This function assigns a numerical value to every library for a given module. A positive value indicates an expression level above the cell population average, while a negative value indicates the opposite. Shapiro-Wilk normality test, or for cases with n>5000, the D'Agostino Skewness and Kurtosis Normality Tests from the 'burrm/lolcat' R package were applied. Next, either Levene's Test or Fligner-Killeen tests of homogeneity of variances were used. Because data was neither normally distributed nor homoscedastic (homogeneity of variances), both the Kruskal-Wallis rank sum test and Welch's ANOVA (One-way analysis of means, not assuming equal variances) were applied. Both tests rejected the null hypothesis, indicating that there is significant difference among groups. Finally, the Conover-Iman test, with Benjamini-Hochberg p adjustment method, for multiple pairwise comparisons across dopaminergic clusters, neighborhoods and territories was applied.

## Correlation between cell loss and vulnerability module score

Two linear regression models were fitted using normalized cell loss as the response variable and vulnerability module either with or without DAT (*Slc6a3*) as the predictor. Since the vulnerability module score was calculated, based on the vulnerable clusters, only the non-vulnerable clusters were used in the model. The module scores were square root (sqrt) transformed to make their distribution normal, prior to fitting. The plots show the $R^2$, the estimated model equations and the 95% confidence interval limits.

## Differential gene expression analysis

Differentially expressed genes between two or more groups were identified using Wilcoxon Rank Sum test in the Seurat FindMarkers or FindAllMarkers functions with the following set parameters; logfc.threshold=0.25, min.pct=0.1, min.diff.pct = -Inf, only.pos=FALSE, max.cells.per.ident=Inf. The calculated p-values were adjusted for multiple comparisons using the Bonferroni correction.

## GO enrichment analysis

The ML territory enriched genes (806 upregulated and 661 downregulated) were used for this analysis by removing the overlapping genes enriched in the DE between all lesioned and all intact (*Figure 7—figure supplement 1*). The 'enrichR' package as the R interface to the Enrichr database, version 3.1 was used with GO_Biological_Process_2021 as the imported gene-set library (*Kuleshov et al., 2016*). The top 20 enriched terms, ranked by p value, were selected per group for visualization.

## High dimensional weighted gene correlation network analysis

The R package hdWGCNA (version '0.2.20') was used (*Langfelder and Horvath, 2008*; *Morabito et al., 2020*; *Morabito et al., 2023*). The dopaminergic neuron subset of the data (mDA) was set up for analysis, using 'fraction' method for gene selection. Prior to the construction of the Metacells, harmony was run on PCA, and then Metacells were constructed, using harmonized PCs as the reduced dimension (reduction = 'harmony'). Metacells were constructed based on the 'territory x condition' grouping. For the co-expression network analysis, the gene expression matrix was set up with the IDs for 'TER.name_lesioned' provided to the 'group_name' argument of the SetDatExpr() function. So, only lesioned cells of the mDA territories were provided, while ML clusters and non-mDA groups were excluded. Next, the soft-power threshold was selected, using a 'signed' network type, with a scale-free Topology Model Fit greater ≥0.8. Next, a co-expression network was constructed, and 9 co-expression modules were calculated. Then, module eigengenes (MEs) and module connectivity (kMEs) were calculated, using the 'group.by' and 'group_name' arguments stated above. Hub genes were selected based on the top 30 ranked kME values. Correlations among modules were calculated using ME as 'feature' and 'sig.level'=0.05. Enrichment analysis on the hdWGCNA modules was done using the R packages 'enrichR' and 'GeneOverlap'. The enrichR databases used were 'GO_Biological_Process_2021', GO_Cellular_Component_2021', and 'GO_Molecular_Function_2021', with maximum number of genes to test per module set to 100, ranked by kME values.

## Generation of human dataset

Human dataset including count matrix, barcodes, features and metadata files were downloaded from the Gene Expression Omnibus (GEO) under the accession number (GSE178265; *Kamath et al., 2022*). Genes with reads present in fewer than 1 nucleus (libraries), and *Malat1* were removed. Libraries with detected UMIs below 650 and UMI/Gene below 1.2 and, with percent_mito above 10 were also removed. Reciprocal PCA (RPCA) approach was used to integrate individual SNc samples separately per condition (control vs diseased), by which each sample is projected into the others' PCA space, and the integration anchors are defined by the requirements of mutual neighborhood. Next the control and diseased dataset were integrated into one dataset, using RPCA approach. Louvain resolution 0.1 was used for clustering and the clusters of dopaminergic nuclei were identified by plotting typical markers. These dopaminergic clusters of 25003 nuclei (control = 17039,, LBD = 4642, PD = 3,322 nuclei) were subset for downstream analysis.

## Integration of human and mouse dataset

Orthology mapping package 'Orthology.eg.db', version 3.13.0 was used to map NCBI gene IDs between species. Human genes were converted to their mouse orthologs and the ones without a mouse ortholog, were also kept. Data from human dopaminergic nuclei, subset in the previous step (above), containing 18 donors (8 control, 6 PD, 4 LBD) and data of the 12 mouse samples from 6 dissected midbrains (6 intact, 6 lesioned) from the dopaminergic dataset ($Th^+$/ $Slc6a3^+$), was first log-normalized, highly variable genes found (method = vst) separately per species, prior to being integrated using the Seurat canonical correlation analysis (CCA).

## Mapping and annotating human control nuclei query dataset

Unimodal UMAP Projection approach from Seurat v4. was used to compute a UMAP reference from the mouse intact, mDA territories nuclei, using the first 30 PCs after log-normalization, finding highly variable genes (method = vst) and scaling of the mouse reference data. Then human control dopaminergic nuclei were used as query to be projected onto the reference UMAP structure, using the MapQuery() function. Next, the mouse intact only *Sox6* territory nuclei were used to make the reference UMAP, with the same parameters as above and the human nuclei with the transferred label 'Sox6' from the previous step were used as a subset query to be mapped with a higher resolution to the *Sox6* neighborhoods (*Figure 8—figure supplement 1*).

## Acknowledgements

This work was supported by Swedish Research Council (VR 2020-00884; VR 2016-02506), The Swedish Brain Foundation (Hjärnfonden), and Torsten Söderbergs Foundation to TP, and a postdoctoral grant

from Karolinska Institutet's Strategic Research Area Neuroscience (StratNeuro, Karolinska Institutet) to BYS. The computational analyses, data handling, and data curation were enabled by resources provided by the National Academic Infrastructure for Supercomputing in Sweden (NAISS) in the project NAISS 2023/23-181 and the Swedish National Infrastructure for Computing (SNIC) in the project SNIC 2022/23-171 at UPPMAX, Uppsala University, partially funded by the Swedish Research Council through grant agreements no. 2022-06725 and no. 2018-05973. The authors wish to acknowledge support from the National Genomics Infrastructure in Stockholm funded by Science for Life Laboratory, the Knut and Alice Wallenberg Foundation and the Swedish Research Council, for assistance with massively parallel sequencing as well as SciLifeLab In Situ Sequencing Infrastructure Units. We are grateful for support from the Stellenbosch Institute for Advanced Studies (STIAS) during the final stage of this project. We also would like to thank Petter Woll, at the Center for Hematology and Regenerative Medicine (HERM) in Neo, Karolinska Institutet, for advice.

## Additional information

### Funding

| Funder | Grant reference number | Author |
|---|---|---|
| Vetenskapsrådet | VR 2020-00884 | Thomas Perlmann |
| Hjärnfonden | | Thomas Perlmann |
| Torsten Söderbergs Stiftelse | | Thomas Perlmann |
| Karolinska Institutet | Postdoctoral grant | Behzad Yaghmaeian Salmani |
| Vetenskapsrådet | VR 2016-02506 | Thomas Perlmann |

The funders had no role in study design, data collection and interpretation, or the decision to submit the work for publication.

### Author contributions

Behzad Yaghmaeian Salmani, Conceptualization, Data curation, Software, Formal analysis, Funding acquisition, Validation, Investigation, Visualization, Methodology, Writing – original draft, Project administration, Writing – review and editing, Bioinformatic analysis; Laura Lahti, Conceptualization, Data curation, Formal analysis, Validation, Investigation, Visualization, Methodology, Writing – original draft, Project administration, Writing – review and editing; Linda Gillberg, Ioannis Mantas, Data curation, Formal analysis, Validation, Investigation, Visualization, Methodology, Writing – original draft, Writing – review and editing; Jesper Kjaer Jacobsen, Formal analysis, Validation, Investigation, Visualization, Methodology, Writing – review and editing; Per Svenningsson, Conceptualization, Resources, Supervision, Investigation, Project administration, Writing – review and editing; Thomas Perlmann, Conceptualization, Resources, Formal analysis, Supervision, Funding acquisition, Validation, Investigation, Visualization, Methodology, Writing – original draft, Project administration, Writing – review and editing

### Author ORCIDs

Behzad Yaghmaeian Salmani ⓘ https://orcid.org/0000-0002-4221-6243
Thomas Perlmann ⓘ http://orcid.org/0000-0003-4821-8036

### Ethics

All experimental procedures in this study followed the guidelines and recommendations of Swedish animal protection legislation and were approved by Stockholm North Animal Ethics board (animal permit number: 13830/18).

Reviewer #1 (Public Review): https://doi.org/10.7554/eLife.89482.3.sa1
Reviewer #2 (Public Review): https://doi.org/10.7554/eLife.89482.3.sa2
Author response https://doi.org/10.7554/eLife.89482.3.sa3

# Additional files

### Supplementary files
• Supplementary file 1. Sample information for the "all-nuclei" and well as the "intact-untreated" datasets in this study. Cluster markers for the "all-nuclei" dataset.

• Supplementary file 2. Cluster markers for the dopaminergic "mDA" dataset.

• Supplementary file 3. The list of primary and secondary antibodies used in this study as well as the ML-clusters specific ISH probes Clic4, Creb5, Mmp12 and Sprr1a. The list of Allen Mouse Brain Atlas experiments of which images were used in this study.

• Supplementary file 4. Lists of padlock probes for in situ RNA sequencing (ISS) and fluorescently labeled detection oligos used in this study.

• Supplementary file 5. Calculated normalized cell loss (see methods) and statistical comparisons of normalized cell loss across clusters, neighborhoods, and territories.

• Supplementary file 6. Identification of vulnerability and resilience modules' genes. Statistical comparisons of both vulnerability and resilience module scores across clusters, neighborhoods and territories.

• Supplementary file 7. Territory membership and annotations of clusters in integrated mDA dataset (mergedTERs_integCLUSTERs). Statistical comparisons of both vulnerability and resilience module scores across integrated mDA clusters.

• Supplementary file 8. Differential gene expression analyses between (1) lesioned vs. intact, (2) ML vs. rest of the population, and (3) untreated vs. intact.

• Supplementary file 9. Gene set enrichment analysis for nine WGCNA co-expression modules (LmDAmod 1-9).

• Supplementary file 10. Gene set enrichment analysis of the genes uniquely dysregulated in ML clusters.

• MDAR checklist

### Data availability
Sequencing Fastq files, raw and normalized count matrixes, and supplementary files are accessible via GEO accession number GSE233866. All-nuclei and mDA-nuclei datasets are accessible as CELLx-GENE interactive web-based tools for visualization and exploration via the links to our lab homepage: https://perlmannlab.org/resources/. The code for analyses and figures in this study is available on GitHub (copy archived at *Yaghmaeian-Salmani, 2024*).

The following dataset was generated:

| Author(s) | Year | Dataset title | Dataset URL | Database and Identifier |
|---|---|---|---|---|
| Yaghmaeian Salmani B, Lahti L, Gillberg L, Jacobsen JK, Mantas I, Svenningsson P, Perlmann T | 2023 | Transcriptomic atlas of midbrain dopamine neurons uncovers differential vulnerability in a Parkinsonism lesion model | https://www.ncbi.nlm.nih.gov/geo/query/acc.cgi?acc=GSE233866 | NCBI Gene Expression Omnibus, GSE233866 |

The following previously published dataset was used:

| Author(s) | Year | Dataset title | Dataset URL | Database and Identifier |
|---|---|---|---|---|
| Kamath T, Abdul A, Burris SJ, Nadaf N, Gazestani V, Vanderburg C, Macosko E | 2021 | A molecular census of midbrain dopaminergic neurons in Parkinsons disease | https://www.ncbi.nlm.nih.gov/geo/query/acc.cgi?acc=GSE178265 | NCBI Gene Expression Omnibus, GSE178265 |

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

## Appendix 1

## Supplementary Results related to *Figure 4—figure supplements 1–3*

### *Sox6* territory

Generally, SNc and its subdomains belonged to the *Sox6* territory, with the exception of SNcL which contained few, if any, *Sox6*$^+$ cells, and which belonged to the *Pdia5* territory (see below). *Sox6* is known to be expressed in the adult ventral SNc, but also in some cells of the VTA (*Panman et al., 2014*). The most ventral *Sox6*$^+$ neurons in SNc co-express *Aldh1a1*, whereas the dorsal tier contains *Aldh1a1*$^-$ *Sox6*$^+$ cells (*Panman et al., 2014*; *Pereira Luppi et al., 2021*). Similarly, in our dataset the four neighborhoods of *Sox6* territory could be divided into two groups based on *Aldh1a1* expression (*Figure 4B*, *Figure 4—figure supplement 1*): NH1 and NH4, which were entirely *Aldh1a1*$^+$, and NH2 and NH3, which were mostly *Aldh1a1*$^-$. *Sox6*$^+$ *Aldh1 a1*$^+$ cells have been reported to express *Anxa1*, *Aldh1a7*, and *Lmo3* and project mainly to dorsolateral caudate putamen (*Poulin et al., 2014*; *Poulin et al., 2018*; *Poulin et al., 2020*; *La Manno et al., 2016*; *Hook et al., 2018*; *Saunders et al., 2018*; *Azcorra et al., 2023*). Although *Lmo3* was found in all *Sox6* neighborhoods, it was indeed most enriched in both NH1 and NH4 (*Figure 3—figure supplement 1*).

NH1 consisted mostly of *Anxa1*$^+$ neurons, localizing in the SNcM and SNcV (*Figure 4—figure supplement 1C, G*). *Anxa1* was also found in NH4 which, based on the expression of *Aldh1a7*$^+$ and *Grin2c*$^+$, most likely matches the most ventral tier of anterior SNc (*Figure 4—figure supplement 1D, G*). The *Anxa1*$^+$ SNpc neurons were recently characterized in a series of behavioral experiments and appear to respond specifically to acceleration (*Azcorra et al., 2023*). However, as so many genes were shared between NH1 and NH4, the exact location of these neighborhoods in the tissue can only be estimated at this point. More definitive localization will require further analyses.

*Sox6*$^+$ *Aldh1a1*$^-$ NH2 enriched markers included *Col11a1* and *Vcan*, although *Vcan* was also found in *Aldh1a1*$^+$ *Sox6*$^+$ cells of NH1, supporting earlier findings (*Saunders et al., 2018*; *Poulin et al., 2020*). Allen Brain Atlas expression data for these genes showed localization mostly in the anterior SNc, corresponding to upper, *Aldh1a1*$^-$ tier (*Figure 4—figure supplement 1E, G*). A small population of NH2 cells express also low levels of *Calb1*, which is more highly expressed in VTA (see below). Projections of these *Aldh1a1*$^-$ *Calb1*$^+$ *Sox6*$^+$ neurons in SNpc are biased towards ventromedial regions of caudate putamen (*Poulin et al., 2018*).

NH3 was mostly *Aldh1a1*$^-$ *Ndnf*$^+$ *Vill*$^+$, although both *Vill* and *Ndnf* were also found to some extent in NH1 and NH4 (*Figure 4—figure supplement 1F, G*). Our ISS sequencing and Allen Brain Atlas data showed that *Ndnf*$^+$ *Aldh1a1*$^-$ neurons in more dorsal tier of SNc extended through the midbrain, and *Vill* in Allen Brain Atlas was detected in the same areas, although in fewer cells. NH3 expressed also *Igf1*, shown to be expressed in both anterior and caudal SNc (*Figure 3—figure supplement 1*, *Pristerà et al., 2019*). Furthermore, this neighborhood expressed *Tacr3* (*Figure 3C*, *Figure 3—figure supplement 1*), which together with *Ndnf* had been reported to be expressed in *Sox6*$^+$ *Aldh1a1*$^-$ neurons which project to striatum (*Poulin et al., 2014*; *Poulin et al., 2018*; *Poulin et al., 2020*), although neurons with similar expression profile were found also in *Pdia5* territory (see below).

One cluster in NH3 did express *Aldh1a1* (*Figure 3C*), which likely represents *Aldh1a1*$^+$ cells detected in more caudal SNc, where neither NH1 nor NH4 -specific markers, such as *Aldh1a7* or *Anxa1*, were found (*Figure 4—figure supplement 1B, D*). Some of these neurons likely correspond to *Sox6*$^+$ + projecting to the core and lateral shell of nucleus accumbens (*Poulin et al., 2018*).

### *Ebf1* territory

*Ebf1*, the territory with fewest cells, consisted of two neighborhoods (*Figure 4A and F*, *Figure 4—figure supplement 2A-C*). NH1, specified by *Col24a1*, was found in scattered cells of PAG, RLi, and PBP, although here the ISS signal was diffuse and hard to interpret (*Figure 4—figure supplement 2A*). This neighborhood, which expressed also *Slc17a6*, *Calb1*, *Cck*, *Npw* and *Man1a*, likely corresponds to a specific population of *Npw*$^+$ DA neurons identified in PAG, VTA and dorsal raphe, which project to central extended amygdala and modulate mouse behavior under stress (*Motoike et al., 2016*).

NH2 expressed *Vip* and these cells were found mostly in periaqueductal grey (PAG) and DR, and to some extent in PBP, PN, RLi and CLi (*Figure 4—figure supplement 2B*). This neighborhood corresponds to the previously identified *Slc17a6*$^+$ *Vip*$^+$ subtype (*Dougalis et al., 2012*; *Poulin et al.,*

*2014*; *La Manno et al., 2016*; *Hook et al., 2018*; *Tiklová et al., 2019*). Vip-Cre- labelled neurons project to lateral part of the central amygdala (*Poulin et al., 2018*).

### *Pdia5* territory

Many of the cells in *Pdia5* territory were also $Sox6^+$ (*Figure 3C*), and may correspond to the $Sox6^+$ $Aldh1a1^-$ subtype that has been detected in PBP and in the most dorsal tier of SNc (*Poulin et al., 2014*; *La Manno et al., 2016*) expressing $Igf1^+$, $Ndnf^+$ and $Tacr3^+$. Based on these markers, it matches *Pdia5* NH1 (*Figure 3*, *Figure 3—figure supplement 1*). *Pdia5* itself showed an interesting expression pattern in VTA, forming a distinct dorsal tier of PBP which extended laterally as a narrow strip of cells above SNc, divided by medial lemniscus (*Figure 4C and F*). A careful chemoarchitectonic and cell morphological analyses of midbrain DA neurons by Paxinos and colleagues assign this dorsal-most tier as a part of PBP, not as a part of SNc proper (*Fu et al., 2012*). Furthermore, the $Sox6^+$ + in this layer project to ventromedial parts of striatum, as opposed to the rest of SNc which mainly projects to dorsolateral striatum (*Pereira Luppi et al., 2021*).

Although *Pdia5* NH1 and NH2 shared many genes, the expression of *Postn* and *Npy1r* in NH2 enabled us to map the majority of cells in this neighborhood to the most lateral tip of SNcL (*Figure 4—figure supplement 2D, E*), which was also $Calb1^+$ (*Figure 4—figure supplement 2F*). In addition, this neighborhood expressed *Slc17a6, Cbln1*, and *Wnt7b* (*Figure 3C*, *Figure 3—figure supplement 1*) and likely represents some of $Slc17a6^+$mDA neurons described earlier (*Poulin et al., 2020*). These neurons innervate mainly the central nucleus of amygdala and the tail of caudate putamen (*Poulin et al., 2018*).

### *Otx2* territory

$Otx2^+$ $Aldh1$ $a1^+$DA neurons in VTA have been described before (*Di Salvio et al., 2010*). In addition to *Otx2*, the previously identified enriched genes for this subtype include *Lpl, Tacr3, Grp*, and *Cbln4* (*Poulin et al., 2014*; *La Manno et al., 2016*; *Tiklová et al., 2019*, *Hook et al., 2018*; *Saunders et al., 2018*). We found these genes to be enriched in the *Otx2* territory, and based on our own analyses and Allen Mouse Brain Atlas data, the cells of this territory appears to be localized in the most medial regions of VTA: IF, PN, medioventral PBP, as well is to some extent in RLi and CLi (*Figure 4D and F*, *Figure 4—figure supplement 2G*), which supported the earlier analyses.

*Otx2* NH1 was $Tacr3^+$ $Aldh1$ $a1^+$ $Slc6a3^{high}$ and also expressed *Grp* and *NeuroD6*, which specify a distinct VTA subtype (*Kramer et al., 2018*). $Grp^+$ $NeuroD6$ + cells innervate the medial shell of the nucleus accumbens, whereas $Grp^+$ $NeuroD6^-$ cells send projections to the dorsomedial striatum (*Kramer et al., 2018*). Aldh1a1-Cre labelled cells in the VTA, corresponding to *Otx2* NH1, have been shown to innervate the medial shell of nucleus accumbens and olfactory tubercle (*Poulin et al., 2018*). NH1 cells were found in PN, PIF and the most medial parts of PBP (*Figure 4—figure supplement 2H*). The $Aldh1a1^-$ $Slc6a3^{low}$ NH2 cells, which also expressed *Csf2rb2*, were mostly localized in RLi and IF (*Figure 4—figure supplement 2I*). *Slc6a7 (Vglut2)* was expressed in both NH1 and NH2, although being more highly expressed in NH2. Vglut2-Cre-labelled $Otx2^+$ + in ventromedial VTA innervate mainly nucleus accumbens medial shell and in discrete areas of olfactory tubercle (*Poulin et al., 2018*). It should be noted that *Slc6a7* is also found in *Ebf1* and *Gad2* territories as well as in *Pdia5* NH2. However, this neighborhood also expresses *Cck* – which is also found in *Ebf1* territory, *Pcsk6* NH2, and *Pdia5* NH1 – and Cck-Cre-labelled neurons in VTA project to basolateral amygdala (*Poulin et al., 2018*).

### *Gad2* territory

$Slc32a1^+$ $Slc17$ $a6^+$ $Slc6a3^{low}$ $Th^{low}$ DA neurons in VTA have been identified in several studies (*Poulin et al., 2014*; *La Manno et al., 2016*; *Tiklová et al., 2019*, *Saunders et al., 2018*), and based on the enriched gene expression match our *Gad2* territory. In support of these earlier studies, we localized these neurons mostly to IF to some extent RLi, CLi and PBP (*Figure 4A and F*, *Figure 4—figure supplement 3A-C*). Both $Ebf2^+$ +1 and $Met$ $^+$NH2 were intermingled in these areas, although NH2 cells concentrated more in the ventral midline of RLi and were less frequent in more caudal VTA (*Figure 4—figure supplement 3C*). These neurons likely correspond to previously described VTA neurons that project to the lateral habenula and function to promote award (*Stamatakis et al., 2013*). Although the majority of neurons in this territory express low levels of both *Th* and *Slc6a3*, they do express the typical mDA neuron signature, suggesting a potential for GABA and DA co-release.

### *Fbn2* territory

The *Fbn2* territory consisted of two small neighborhoods, both of which were localized in the antero-medial PBP (*Figure 4A and F*, *Figure 4—figure supplement 3D-G*). In addition to the most specific territory marker *Fbn2*, these cells expressed other less specific identifiers such as *Slc17a6, Mkx, Mid1, Glra3* and *Otx2* (*Figure 3*, *Figure 3—figure supplement 1*) and some of them likely corresponded to $Slc17a6^+$ mDA subtype in VTA (*Poulin et al., 2020*; *Garritsen et al., 2023*).

NH1 was further specified by *Rxfp1*, whereas NH2 expressed *Dsg2* and *Col23a1* (*Figure 4—figure supplement 3F, G*). Another highly enriched, although not exclusive, expressed gene in *Fbn2* territory was *Bcl11a* (*Figure 3—figure supplement 1*), suggesting that some of the recently identified $Bcl11a^+$ DA neurons in VTA (*Tolve et al., 2021*) likely belong to this territory. Some of the neurons in *Fbn2*-territory likely match to Vglut2-labelled cells in PBP projecting to nucleus accumbens (*Poulin et al., 2018*).

### *Pcsk6* territory

*Pcsk6* territory expressed *Ano2* but had very low *Slc17a6*. Pcsk6_NH1 expressed *Ranbp3l, Cgnl1, Cpne2, Ccdc192 (1700011I03Rik)* and *Kank1* (*Figure 3—figure supplement 1*, *Figure 4—figure supplement 3H*), and did not correspond well to any mDA neuron types described previously. Our ISS analysis discovered some of $Ccdc192^+$ cells in PBP where a less exclusive NH1 marker *Cpne2* was also found (*Figure 4—figure supplement 3I*).

Pcsk6_NH2 was best identified by the expression of *Tacr3* and the lack of *Aldh1a1* (*Figure 4E*). Some of them, which also express *Sox6* and *Ndnf*, match previously described $Sox6^+$ $Aldh1a1^-$ $Ndnf^+$ DA 1B -neurons (*Poulin et al., 2014*), $Cyp26b1^+$ $Sox6^+$ $Tacr3^+$ mDA subtype 4–3 neurons (*Saunders et al., 2018*; *Poulin et al., 2020*) and $Cd9^+$ $Ntf3^+$ DA-VTA1 neurons (*La Manno et al., 2016*). These cells were found in PBP, IF, PN, CLi, and RRF (*Figure 4F*). Together with *Pdia5* NH1, they formed a dorsolateral extension of PBP.

