## [Editor Report · eLife assessment]

This **important** study investigated transcriptional profiles of midbrain dopamine neurons using single nucleus RNA (snRNA) sequencing. The authors found more nuanced subgroups of dopamine neurons than previous studies, and identified some genes that are preferentially expressed in subpopulations that are more vulnerable to neurochemical lesions using 6-hydroxydopamine (6OHDA). The results are **convincing** and provide critical information on the heterogeneity and vulnerability of dopamine neurons which will be a foundation for future studies.

---

## [Referee Report · Reviewer #1 (Public Review)]

In this study by Yaghmaeian Salmani et al., the authors performed single-nuclei RNA sequencing of a large number of cells (>70,000) in the ventral midbrain. The authors focused on cells in the ventral tegmental area (VTA) and substantia nigra (SN), which contain heterogeneous cell populations comprising dopaminergic, GABAergic, and glutamatergic neurons. Dopamine neurons are known to consist of heterogeneous subtypes, and these cells have been implicated in various neuropsychiatric diseases. Thus, identifying specific marker genes across different dopamine subpopulations may allow researchers in future studies to develop dopamine subtype-specific targeting strategies that could have substantial translational implications for developing more specific therapies for neuropsychiatric diseases.

A strength of the authors' approach compared to previous work is that a large number of cells were sequenced, which was achieved using snRNA-seq, which the authors found to be superior compared to scRNA-seq for reducing sampling bias. A weakness of the study is that relatively little new information is provided as the results are largely consistent with previous studies (e.g., Poulin et al., 2014). Nevertheless, it should be noted that the authors found some more nuanced subdivisions in several genetically identified DA subtypes.

Lastly, the authors performed molecular analysis of ventral midbrain cells in response to 6-OHDA exposure, which leads to the degeneration of SN dopamine neurons, whereas VTA dopamine neurons are largely unaffected. Based on this analysis, the authors identified several candidate genes that may be linked to neuronal vulnerability or resilience.

Overall, the authors present a comprehensive mouse brain atlas detailing gene expression profiles of ventral midbrain cell populations, which will be important to guide future studies that focus on understanding dopamine heterogeneity in health and disease.

Comments on the revised version

The authors have addressed all of my concerns.

---

## [Referee Report · Reviewer #2 (Public Review)]

In the manuscript by Salmani et al., the authors explore the transcriptomic characterization of dopamine neurons in order to explore which neurons are particularly vulnerable to 6-OHDA-induced toxicity. To do this they perform single nucleus RNA sequencing of a large number of cells in the mouse midbrain in control animals and those exposed to 6-OHDA. This manuscript provides a detailed atlas of the transcriptome of various types of ventral midbrain cells - though the focus here is on dopaminergic cells, the data can be mined by other groups interested in other cell types as well. The results in terms of cell type classification are largely consistent with previous studies, though a more nuanced picture of cellular subtypes is portrayed here, a unique advantage of the large dataset obtained. The major advance here is exploring the transcriptional profile in the ventral midbrain of animals treated with 6-OHDA, highlighting potential candidate genes that may influence vulnerability. This approach could be generalizable to investigate how various experiences and insults alter unique cell subtypes in the midbrain, providing valuable information about how these stimuli impact DA cell biology and which cells may be the most strongly affected.

Comments on the revised version

The authors addressed most of my concerns about the depth of analysis and implemented further analyses of the data. However I still think that the manuscript would be strengthened with an acknowledgement and deeper integration with the concepts from recent papers in the field, as mentioned by Reviewer 1. There is a rich amount of biology that can be gleaned from understanding the anatomical topology of the VTA and how that relates to gene expression patterns, both at a basal state and following 6-OHDA injection. For example, I made the point about medially-located DA cells in the VTA being the DA that co-express vGluT2. The work would provide more value to the field if more effort was made in the introduction and discussion to briefly mention the recent key papers in the field and how their work relates to our knowledge of the VTA and adjacent SNc in terms of cell-type identity, spatial location, and co-expression of various genes e.g., DAT and vGluT2.

---

## [Author Response]

The following is the authors’ response to the original reviews.

**eLife assessment**
This study investigated transcriptional profiles of midbrain dopamine neurons using single nucleus RNA (snRNA) sequencing. The authors found more nuanced subgroups of dopamine neurons than previous studies, and idenfied some genes that are preferenally expressed in subpopulaons that are more vulnerable to neurochemical lesions using 6-hydroxydopamine (6OHDA). The reviewers found the results are solid, and the study is overall valuable, providing crical informaon on the heterogeneity and vulnerability of dopamine neurons although the scope is somewhat limited because the result with snRNA is similar to previous results and cell deaths were induced by 6OHDA injecons.
**Public Reviews:**

**Reviewer #1 (Public Review):**
In this study by Yaghmaeian Salmani et al., the authors performed single-nuclei RNA sequencing of a large number of cells (>70,000) in the ventral midbrain. The authors focused on cells in the ventral tegmental area (VTA) and substana nigra (SN), which contain heterogeneous cell populaons comprising dopaminergic, GABAergic, and glutamatergic neurons. Dopamine neurons are known to consist of heterogeneous subtypes, and these cells have been implicated in various neuropsychiatric diseases. Thus, idenfying specific marker genes across different dopamine subpopulaons may allow researchers in future studies to develop dopamine subtype-specific targeng strategies that could have substanal translaonal implicaons for developing more specific therapies for neuropsychiatric diseases.A strength of the authors' approach compared to previous work is that a large number of cells were sequenced, which was achieved using snRNA-seq, which the authors found to be superior compared to scRNA-seq for reducing sampling bias. A weakness of the study is that relavely litle new informaon is provided as the results are largely consistent with previous studies (e.g., Poulin et al., 2014). Nevertheless, it should be noted that the authors found some more nuanced subdivisions in several genecally idenfied DA subtypes.

On this point we respectfully disagree with the reviewer. In this study, over 30,000 mDA neurons have been analyzed at the genome-wide gene expression level, idenfying mDA territories and neighborhoods (that some may call “subtypes”), a descripon of the mDA neuron diversity that goes far beyond what has been published previously.

Although several single-cell RNA sequencing studies of mDA neurons have added to our understanding of mDA diversity, they have been limited by the low numbers of sequenced mDA neurons. As the reviewer specifically referred to the study by Poulin et al., 2014, it should be noted that in this report, 159 mDA neurons were analyzed by qPCR – not by RNAseq – of 96 previously identified marker genes. Despite those limitaons, this was indeed a highly impressive study, suggesng five different mDA neuron subtypes (as compared to the 16 neighborhoods described here), published before the era of single-cell genome-wide gene expression methods and advanced bioinformac tools were available. On average, the following scRNAseq studies typically captured a few hundred mDA neurons - compared to over 30,000 in this study. None of the studies menoned in our manuscript were close to capturing the full diversity, and the informaon on mDA neuron diversity is, for this reason, somewhat fragmented in the scienfic literature. Indeed, the seven mDA “subtypes” described in the excellent reviews by Poulin et al., 2020 in Trends in Neurosciences and Garritsen et al., 2023 in Nature Neuroscience are integrated interpretaons of the results from numerous independent studies, each methodologically unique. Several previously idenfied groups, especially Vglut2+ populaons in VTA and SNpc, have been considered poorly defined. As menoned above, our findings in this study could reliably idenfy, by computaonal analyses and combinatorial marker expression in situ, 16 different neighborhoods within the mDA populaon and localize them in the ssue (Figure 4, Supplementary figures 4-1 to 4-3, described further in Supplementary Results). To menon three examples: Within Sox6+ SNpc, we idenfied four different variants (neighborhoods) with partly unique anatomical localizaon. In addion, the large group of mDA neurons referred to as the Pcsk6 territory has not been clearly defined in earlier studies. We also idenfied a novel mDA neuron group that is related to the previously well described Vip-expressing mDA neurons. These and other novel features are menoned in the manuscript and in Supplementary Figure 4-1 to 4-3.

Although we have, for the consideraon of the space and intelligibility, characterized the 16 neighborhoods with only a few selected key marker genes, we have idenfied numerous addional novel markers, some of which are shown in dot plots in Figure 3 and Supplementary Figure 3, which can be used to characterize these groups further. We also provide all our sequencing data and our Padlock probe ISS data for anyone to download and analyze further, and we have made a web-based tool, CELLxGENE, available on our group’s website to facilitate exploraon of the different aspects of our dataset.

Lastly, the authors performed molecular analysis of ventral midbrain cells in response to 6-OHDA exposure, which leads to the degeneraon of SN dopamine neurons, whereas VTA dopamine neurons are mainly unaffected. Based on this analysis, the authors idenfied several candidate genes that may be linked to neuronal vulnerability or resilience.Overall, the authors present a comprehensive mouse brain atlas detailing gene expression profiles of ventral midbrain cell populaons, which will be important to guide future studies that focus on understanding dopamine heterogeneity in health and disease.We thank the reviewer for poinng this out.
**Reviewer #2 (Public Review):**
In the manuscript by Salmani et al., the authors explore the transcriptomic characterizaon of dopamine neurons in order to explore which neurons are parcularly vulnerable to 6-OHDA-induced toxicity. To do this they perform single nucleus RNA sequencing of a large number of cells in the mouse midbrain in control animals and those exposed to 6-OHDA. This manuscript provides a detailed atlas of the transcriptome of various types of ventral midbrain cells - though the focus here is on dopaminergic cells, the data can be mined by other groups interested in other cell types as well.The results in terms of cell type classificaon are largely consistent with previous studies, though a more nuanced picture of cellular subtypes is portrayed here, a unique advantage of the large dataset obtained. The major advance here is exploring the transcriponal profile in the ventral midbrain of animals treated with 6-OHDA, highlighng potenal candidate genes that may influence vulnerability. This approach could be generalizable to invesgate how various experiences and insults alter unique cell subtypes in the midbrain, providing valuable informaon about how these smuli impact DA cell biology and which cells may be the most strongly affected.

We appreciate these comments. We want to state that the study not only gives a more nuanced picture but goes far beyond previously published studies and provides a highly resolved and detailed atlas of mDA neurons. Thus, it clarifies poorly described diversity and idenfies enrely novel groups of diverse mDA neurons at the genome-wide gene expression level.

Overall, the manuscript is relavely heavy on characterizaon and comparavely light on funconal interpretaon of findings. This limits the impact of the proposed work. It also isn't clear what the vulnerability factors may be in the neurons that die. Beyond the characterizaon of which neurons die - what is the reason that these neurons are suscepble to lesion? Also, the interpretaon of these findings is going to be limited by the fact that 6-OHDA is an injectable, and the effects depend on the accuracy of injecon targeng and the equal access of the toxin to access all cell populaons. Though the site of injecon (MFB) should hit most/all of the forebrain-projecng DA cells, the injecon sites for each animal were not characterized (and since the cells from animals were pooled, the effects of injecon targeng on the group data would be hard to determine in any case).

We agree that the results are presented to provide a comprehensive and valuable resource rather than explaining molecular mechanisms. The reviewer points out that “what the vulnerability factors may be in the neurons that die” is unclear. However, our study was designed to answer the queson: What genes are enriched in clusters of mDA neurons that are parcularly likely to die aer toxic stress? Using single-cell analysis, we believe this queson had higher priority than atempng to idenfy gene expression changes occurring during the cell death process. We agree that we cannot answer why neurons are suscepble to lesions, only idenfy genes that correlate with either high or low sensivity. Thus, the genes we refer to as “vulnerability genes” and “resilience genes” are candidates for influencing differenal vulnerability. Hard evidence for such influence will require addional and extensive funconal analysis. As for the variability of injecon and the characterizaon of individual animals, we wish to menon the online interacve explorer available at https://perlmannlab.org/resources/ . It allows visualizaon of nuclei distribuon per territory and neighborhood for each mouse, making it easy to determine the cell loss rao and cell distribuon per animal. There is indeed variance in the proporons of intact/lesioned total nuclei per animal. This is also evident from the DAT autoradiographs shown for each lesioned animal and presented in Figure Supplement 5-1 A. Importantly, the relave UMAP distribuon of nuclei is quite similar between individual animals. To further invesgate this, we used Pearson’s Chi square test of independence with a conngency table for animals, each with two categorical variables as the proporon of nuclei from intact vs lesioned parts of the vMB (see added Supplementary figure 5-1 C ). This shows that – while there is a difference in the number of nuclei remaining aer lesioning – the relave distribuon among clusters and neighborhoods is similar between animals. We have clarified this point in the manuscript (see page 12 ).

I am also not clear why the authors don't explore more about what the genes/pathways are that differenate these condions and why some cells are parcularly vulnerable or resilient. For example, one could run GO analyses, weighted gene co-expression network analysis, or any one of a number of analysis packages to highlight which genes/pathways may give rise to vulnerability or resilience. Since the manuscript is focused on idenfying cells and gene expression profiles that define vulnerability and resilience, there is much more that could have been done with this based on the data that the authors collected.

We performed GO analysis for the genes upregulated and downregulated in the ML clusters (specific to the lesion condion) in the original manuscript (Please see figure supplement 7-1 C-E, and the newly added Supplementary file 10), but we agree with the reviewer that we could also have analyzed funconal categories of genes correlang with differenal vulnerability. Thus, we have used tools recently developed by Morabito et al., Cell Reports Methods (2023), and their hdWGCNA package to address this queson. This method is parcularly suitable for analyzing high-dimensional transcriptomics data such as single-cell RNA-seq or spaal transcriptomics. We calculated the coexpression network based on the lesioned nuclei of the mDA territories. Of the 9 co-expression modules calculated, one has the highest expression in Sox6 territory and has genes in common with the vulnerability module. Another co-expression module has genes in common with the resilience module and is most highly expressed in Otx2 and Ebf1 territories. We also did GO analysis for these co-expression modules and added addional GO analysis of the ML-enriched genes (see Supplementary Figure 7-1 D,E, the newly added Supplementary Figure 6-3, and the newly added Supplementary file 9). Text describing these addional analyses are menoned on page 15 and 17.

In addition, we wish to emphasize our idenficaon of the genes we refer to as vulnerability and resilience modules in the previous version of the manuscript. Several of the genes were discussed in the previous version of the manuscript but we have now included more informaon on these genes, based on previously published studies and discuss their potenal funconal roles (see pages 22 & 23 in the Discussion).

Another limitation of this study as presented is the missed opportunity to integrate it with the rich literature on midbrain dopamine (and non-dopamine) neuron subtypes. Many subtypes have been explored, with divergent funcons, and can usually be disnguished by either their projecon site, neurotransmiter identy, or both. Unfortunately, the projecon site does not seem to track parcularly well with transcriptomic idenes, aside from a few genes such as DAT or the DRD2 receptor. However, this could have been more thoroughly explored in this manuscript, either by introducing AAVretro barcodes through injecon into downstream brain sites, or through exisng evidence within their sequencing dataset. There are likely clear interpretaons from some of that literature, some of which may be more excing than others. For example, the authors note that vGluT2-expressing cells were part of the resilient territory. This might be because this is expressed in medially-located DA cells and not laterally-located ones, which tends to track which cells die and which don't.

The manuscript consists of a comprehensive descripon of transcriponal diversity. Although of clear value, we believe that addional, comprehensive analysis that combines snRNAseq with, e.g., AAVretro barcodes must be done in a separate study. It should also be noted that we describe each territory and neighborhoods in the further detail in the Supplementary Results, which contains references to the relevant literature. In line with the comments, this secon has now been expanded with further references to relevant studies (see Supplementary Results related to Figure 4-figure supplements 1-3).

It is not immediately clear why the authors used a relaxed gate for mCherry fluorescence in Figure 1.This makes it difficult to definively isolate dopaminergic neurons - or at least, neurons with a DATCre expression history. While the expression of TH/DAT should be able to give a fairly reliable idenficaon of these cells, the reason for this decision is not made clear in the text.

We used a relaxed gang to ensure that we could capture nuclei expressing low levels of RFP, which we believe could be especially relevant for the lesioned dataset (see page 5). We did not find that it would be advantageous to use a more stringent gang that would risk losing all cells expressing no (or very low levels) RFP. Idenfying mDA neurons based on their typical markers is straighorward, as their transcriponal relaonship is evident from the expression profile of several markers, including transcripon factors such as Nr4a2, Pitx3, and En1. In addion, as pointed out in response to Reviewer #1, point 5, atypical DA neurons expressing Th and other mDA markers with no or low levels of Slc6a3 (DAT) were isolated. We believe the study is more complete by the inclusion of these cells. Moreover, we included a sufficiently large number of cells, which ensured a comprehensive analysis of mDA neurons in relaon to other cell types dissected from the ventral midbrain.

**Reviewer #1 (Recommendations For The Authors):**
(1) The authors state that a major advantage of their approach is that it prevents biased datasets when compared to methods that rely on capturing certain cell types. I was wondering if the authors could follow up on this topic with a more detailed descripon of their methodological advantages regarding potenal sampling bias. This is somewhat unclear to me, given that the results of the present study are largely consistent with previous work on this topic.

As expanded on above (see response to the inial comment in the public review), we strongly disagree that there is litle novelty in our study. None of the previous studies come close to describing the mDA neuron populaon with a similar resoluon, which is unsurprising given the differences in the number of analyzed mDA neurons in this versus previous reports. We agree with the reviewer that our data is consistent with previous studies, when they are all combined. Thus, we idenfied mDA neuron groups that correspond (or roughly correspond) to major DA neuron groups idenfied in previous studies (see pages 8-14 in the Supplementary Results). However, the atlas presented here goes well beyond anything published in scope and resoluon. The diversity we define is comparable to findings that, with careful cross-paper analyses, can be stched together from previous single-cell studies. However, even such a combined analysis does not unravel the resoluon and diverse categorizaon of what we have demonstrated herein (16 neighborhoods in midbrain dopaminergic territories). Considering the well-established problems of dissociang and isolang whole neurons from adult brain ssue, this is likely due to sampling bias, resulng in an almost complete exclusion of some sub-populaons of neurons. We have added text on page 20 to clarify this point.

(2) In the abstract, the authors state that their "results showed that differences between mDA neuron group could best be understood as a connuum without sharp differences between subtypes". However, I am not sure whether this is the most appropriate descripon of the authors' results, parcularly when looking at the schemac overview shown in Fig. 4F. To me, it seems more likely that genecally-defined DA subtypes overlap with discrete ventral midbrain subnuclei - parcularly in the case of Sox6-expressing cells, which are almost exclusively located in the SNc. In the case of genes that are specific for the VTA, there also seems to be a strong bias toward certain VTA subnuclei, although I agree that arguments can be made that there is some topographic organizaon along a dorso-ventral and medio-lateral gradient, which seems to be largely consistent with the anatomical locaon of projecon-defined dopamine neurons as described previously by Poulin et al., 2018 (Nature Neuroscience).

What was meant by connuum must be interpreted in the context of the transcriponal landscape of mDA neurons and not their anatomical localizaon. As stated in the paper, the dendrogram depicon of mDA neurons’ transcriptome can be misinterpreted as an indicaon of sharp boundaries and discrete groups in transcriponal profiles. In contrast, we assert that differences between developmentally related mDA neurons are beter described as a connuum with areas in the gene expression landscape defined by the expression of shared genes but without sharp borders between them. We decided to name different areas within this connuum as “territories” at the higher hierarchical level and “neighborhoods” at the more highly resolved level. Hypothecally, such categorizaon can be even more fine-grained, but we find it unlikely that a resoluon beyond the neighborhood level is biologically relevant. As pointed out, the Sox6 territory is the territory that best qualifies as a disncve subtype, while mDA neurons in, e.g., the VTA consist of much higher and nuanced diversity. Importantly, all mDA neurons are much more related to each other than cell types lacking a common developmental origin, including hypothalamic DA neurons. Thus, our effort to define differences in such a gene expression connuum is, in our opinion, more accurate than conveying the message that the diversity consists of subtypes comparable in difference to other cell types that lack a close developmental relaonship with the mDA neuron populaon. Such disnct neuron types, despite using the same neurotransmiter as hypothalamic DA neurons, appear as disnct islands in the UMAP snRNA-seq landscape and typically harbor hundreds of differenally expressed genes. As pointed out in the Discussion, several other studies have noted similar difficules in defining different subtypes among related neurons in e.g. the cortex, striatum, and hippocampus (Kozareva et al., 2021; Saunders et al., 2018; Tasic et al., 2018; Yao et al., 2021). For example, Yao et al., 2021, used a similar hierarchical definion to avoid the implicaon that different groups (“neighborhoods” in this study) should be defined as disnct subtypes of neurons with obvious disncve funcons.

(3) I recommend that the authors revise the introducon to include more current literature on this topic. The review by Bjoerklund and Dunnet, 2006, is very informave and important, but there is more current literature available that discusses anatomical, molecular, and funconal heterogeneity in the ventral midbrain. For example, it would be nice to incorporate recent work from the Awatramani lab on the mapping of the projecon of molecularly defined dopamine neurons (Poulin et al., 2018; Nature Neuroscience).

We deliberately avoided including primary references to previously described diversity in the Introducon since numerous papers are relevant to cite. Instead, we refer to three essenal reviews, including the recent arcles from Awatramani and Pasterkamp. In the Supplementary Results related to Figure 4 (pages 8-14 in the Supplementary Results), we include many references and the Poulin 2018 paper. We believe that this is the appropriate place for a comprehensive discussion on anatomical, molecular, and funconal heterogeneity. In the revised manuscript's main body, we now emphasize that previous literature is discussed in the Supplementary Results (see page 11).

(4) In Fig. 1C, the authors show a sample image demonstrang overlap between TH and mCherry, but this has not been quanfied. Similarly, there seem to be no sample images and quanficaon for the contralateral side that was exposed to 6-OHDA.

The mouse lines used here (Dat-Cre and Rpl10a-mCherry) have been characterized before (Toskas et al., Science Advances 2022). The labelling colocalizes nearly fully with TH, with some excepons (see response below to point #5). We have now complemented with addional data showing an IHC image of one of the midbrain of a unilaterally lesioned mouse in Figure Supplement 5-1E.

(5) The authors state that they focused their analysis on 33,052 nuclei expressing above-threshold levels of either Th OR Slc6a3. However, there seem to be cell populaons in the ventral midbrain of mice that express TH mRNA but not TH protein, and these cells do not seem to be bona fide dopamine neurons (see work from the Morales lab). Similarly, not all dopamine neurons may express DAT mRNA. I was wondering how these discrepancies may influence the authors' analysis and interpretaon.

Indeed, the presence of cells lacking TH protein despite Th mRNA being expressed has been previously described. We also detected these cells across SNpc and VTA and now show these data as a newly added supplementary figure 2-1. In our dataset, the Gad2 territory, located in the ventromedial VTA, contains cells that express many typical mDA markers, such as Pitx3, but very low levels of TH protein. We have idenfied these based on Pitx3-EGFP and Gad2 mRNA co-expression (figure supplement 4-3). In other parts of VTA and SNpc, most cells seem to co-express Th mRNA and protein and are labeled with Dat-Cre. Also scatered in these areas, we could detect some rare mDA cells that lack TH protein. It should be noted that in our mDA territories other typical mDA neuron genes were expressed, such as Slc18a2, Ddc, Nr4a2 and Pitx3, and thus, they were not solely defined by the presence of Th and/or Slc6a3. Cells that do not have a history of DAT-expression, and therefore were not mCherry labelled, were also included in the analysis due to the relaxed gang used during FANS isolaon.

(6) The sex and age of the mice that are used for the experiments are not stated in the Materials and Methods secon under "Mouse lines and genotyping".

Thank you for pointing this out. This informaon has been added to the updated manuscript in the methods secon.

**Reviewer #2 (Recommendations For The Authors):**
I think that the manuscript can be significantly improved just by providing deeper analyses of the exisng data and linking them to the current state of the art in terms of defining midbrain dopamine neurons (e.g., by projecon). The dataset is likely richer than was explored in the manuscript and more valuable insights could be gleaned with a deeper analysis.

Please see our response to Reviewer #2 (Public Review), regarding WGCNA analysis, and the comments on ML-based GO analysis, as well as the comments on the added secons in the supplementary results file.